# BrainOmni: A Brain Foundation Model for Unified EEG and MEG Signals

**Qinfan Xiao**[1,2*]**, Ziyun Cui**[1,2*]**, Chi Zhang**[1,2]**, Siqi Chen**[2]**, Wen Wu**[1]
**Andrew Thwaites**[3,4]**, Alexandra Woolgar**[3,5]**, Bowen Zhou**[1,2]**, Chao Zhang**[2,1,4†]

[1] Shanghai Artificial Intelligence Laboratory, China
[2] Department of Electronic Engineering, Tsinghua University, China
[3] Department of Psychology, University of Cambridge, UK
[4] Speech Hearing and Phonetic Sciences, University College London, UK
[5] MRC Cognition and Brain Sciences Unit, University of Cambridge, UK.
`{xiaoqf22, cui-zy24}@mails.tsinghua.edu.cn, cz277@tsinghua.edu.cn`

## Abstract

Electroencephalography (EEG) and magnetoencephalography (MEG) measure neural activity non-invasively by capturing electromagnetic fields generated by dendritic currents. Although rooted in the same biophysics, EEG and MEG exhibit distinct signal patterns, further complicated by variations in sensor configurations across modalities and recording devices. Existing approaches typically rely on separate, modality- and dataset-specific models, which limits the performance and cross-domain scalability. This paper proposes BrainOmni, the first brain foundation model that generalises across heterogeneous EEG and MEG recordings. To unify diverse data sources, we introduce BrainTokenizer, the first tokenizer that quantises spatiotemporal brain activity into discrete representations. Central to BrainTokenizer is a novel Sensor Encoder that encodes sensor properties such as spatial layout, orientation, and type, enabling compatibility across devices and modalities. Building upon the discrete representations, BrainOmni learns unified semantic embeddings of brain signals by self-supervised pretraining. To the best of our knowledge, it is the first foundation model to support both EEG and MEG signals, as well as the first to incorporate large-scale MEG pretraining. A total of 1,997 hours of EEG and 656 hours of MEG data are curated and standardised from publicly available sources for pretraining. Experiments show that BrainOmni outperforms both existing foundation models and state-of-the-art task-specific models on a range of downstream tasks. It also demonstrates strong generalisation to unseen EEG and MEG devices. Further analysis reveals that joint EEG-MEG (EMEG) training yields consistent improvements across both modalities. Code and checkpoints are publicly available at `https://github.com/OpenTSLab/BrainOmni`.

## 1 Introduction

Neuronal activity underpins human brain function. This activity generates electrical currents in the cortex, which in turn produces secondary electrical and magnetic fields. These fields can be indirectly measured using non-invasive techniques such as electroencephalography (EEG) and magnetoencephalography (MEG) [42]. EEG measures electrical potentials through electrodes placed on the scalp, while MEG uses either gradient or amplitude sensors to measure the magnetic field at specific

---

*Equal contribution.
†Corresponding author.

39th Conference on Neural Information Processing Systems (NeurIPS 2025).

locations and orientations outside the head. As rich sources of neural activity with high temporal resolutions, EEG and MEG have been widely used in applications such as motor imagery [59], emotion recognition [20], multimodal neural decoding [39, 14, 7], and clinical assessments [34, 88, 46, 69]. However, the majority of these applications are developed separately for each domain, often tailored to specific tasks, datasets, or recording setups [71, 86, 34, 26, 22]. Such specialised models suffer from limited generalisation and poor scalability across tasks and domains. Recently, EEG foundation models have emerged to address these limitations by learning general-purpose neural representations from large-scale data [84, 33, 78]. Although MEG signals offer significantly higher spatial resolution than EEG, foundation models for MEG remain largely unexplored, likely due to the modality's complexity and limited data availability. Notably, EEG and MEG share a common biophysical origin, both capturing neural activity through electromagnetic fields generated by dendritic currents. This shared foundation raises a compelling question: can we develop a unified model that learns from both EEG and MEG data to generalise across modalities and enhance performance in each?

Integrating EEG and MEG signals into a unified foundation model faces two key challenges. First, EEG and MEG exhibit distinct signal characteristics and patterns, posing a cross-modality integration challenge. Second, device heterogeneity and lack of standardisation present a significant cross-device generalisation challenge, both within and across EEG and MEG modalities. This heterogeneity includes differences in electrode/sensor configuration (*e.g.*, count, type, position, placement) and naming conventions – particularly pronounced in MEG. MEG sensors can differ in both type (gradiometer *v.s.* magnetometer) and measurement directions (perpendicular *v.s* tangenital), which adds further complexity to unified modelling.

This paper proposes BrainOmni, the first foundation model for unified EEG and MEG signals. The training of BrainOmni consists of two stages: **(i)** unifying heterogeneous data into the same feature space; **(ii)** capturing semantic features of brain activity. In the first stage, we introduce BrainTokenizer. Inspired by source activity estimation [1], BrainTokenizer learns to infer spatiotemporal patterns of brain activity from the observed EEG/MEG signals and generate quantised discrete tokens.

To address device heterogeneity, we propose a novel Sensor Encoder that utilises each sensor's physical characteristics, such as spatial coordinates, orientation, and type, rather than relying solely on channel naming conventions that are often inconsistent across devices and datasets. This design enables BrainTokenizer to handle arbitrary EEG/MEG signal inputs, laying the foundation for large-scale joint pretraining. By integrating EEG/MEG signals with sensor metadata, BrainTokenizer infers a set of *latent source variables* that represent the dominant generative factors underlying electroencephalographic and magnetoencephalographic measurements across diverse sensor configurations, thereby unifying heterogeneous data into a common feature space for downstream modelling.

Building on the discrete brain representations produced by BrainTokenizer, BrainOmni learns rich semantic representations of neural activity through self-supervised pretraining in the second stage. To support this, we curated and standardised a large-scale dataset comprising 1,997 hours of EEG and 656 hours of MEG recordings. Experimental results show that BrainOmni: **(i)** outperforms existing methods across a range of downstream tasks; **(ii)** generalises effectively to previously unseen EEG and MEG devices; and **(iii)** consistently benefits from joint EEG-MEG (EMEG) training across modalities. Our key contributions are as follows:

- **Joint EMEG pretraining.** To the best of our knowledge, BrainOmni is the first single model to perform unified pretraining on both EEG and MEG signals.

- **Modelling physical heterogeneity across devices.** To address the heterogeneity of EEG and MEG recording devices, we propose a novel Sensor Encoder that operates independently of electrode naming conventions or fixed topologies, enabling compatibility across both devices and signal modalities.

- **Spatiotemporal brain signal quantisation.** To our knowledge, BrainTokenizer is the first model that enables spatiotemporal quantisation of brain signals.

## 2 Method

BrainOmni consists of two training stages. In Stage 1, BrainTokenizer is developed to discretise heterogeneous EMEG signals into semantically rich representations. In Stage 2, the BrainOmni

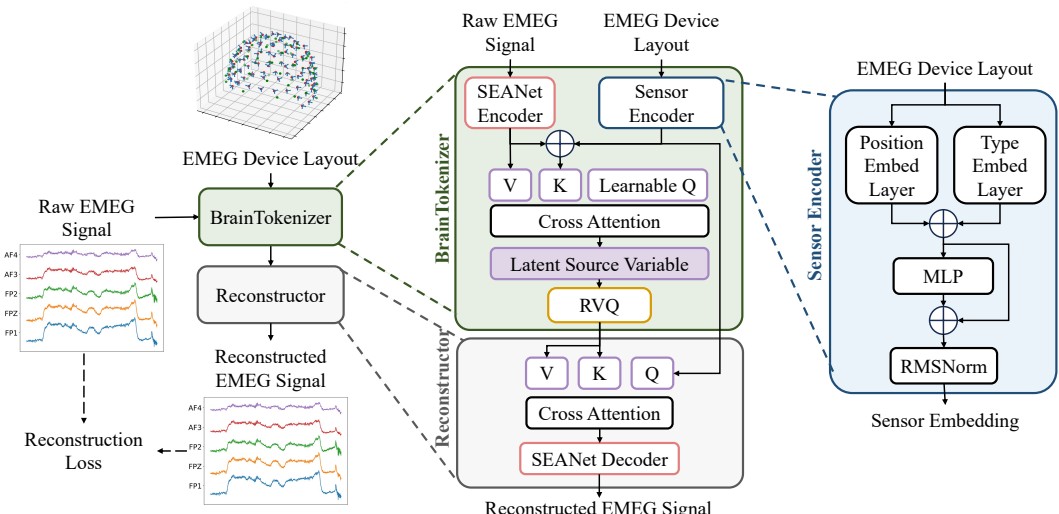

Figure 1: Illustration of the training pipeline of BrainTokenizer. Left: Overview of the autoencoder scheme for BrainTokenizer training. Middle: Structure of the BrainTokenizer and Reconstructor. Right: Structure of the Sensor Encoder.

model is trained based on BrainTokenizer's outputs with a masked token prediction framework, leveraging longer continuous samples to capture extended temporal dependencies.

## 2.1 EMEG Signals

EEG measures electrical potentials through electrodes placed on the scalp, while MEG uses either gradiometer (GRAD) or magnetometer (MAG) sensors to measure the magnetic field at specific locations and orientations outside the head. The measured EMEG signals are multichannel time-series data, which can be represented as $\mathbf{X} \in \mathbb{R}^{C \times T}$, where $C$ denotes the number of sensor channels, and $T$ denotes the number of sampling points. Taking the sensor type, locations and orientations into account, an EMEG sample can be donated as $\mathbf{\Omega} = (\mathbf{X}, \mathbf{L}, \mathbf{S})$, where $\mathbf{X}$ is the raw signal, $\mathbf{L} \in \mathbb{R}^{C \times 6}$ is the physical position and orientation information of each sensor in Cartesian coordinates, and $\mathbf{S} = \{s_1, s_2, \ldots, s_c | s_i \in \{0 \text{ (EEG)}, 1 \text{ (GRAD)}, 2 \text{ (MAG)}\}\}$ is the type of each sensor.

## 2.2 BrainTokenizer

The overall pipeline for Stage 1 is illustrated in Fig. 1. BrainTokenizer is trained using a masked autoencoder scheme [15], where it quantises EMEG samples into compact discrete tokens representing latent features, and a reconstructor module recovers the original EMEG signals from these tokens. Stage 1 is trained using a reconstruction loss between the original and reconstructed signals, which enables the BrainTokenizer to generate quantised latent representations which effectively capture temporal and spatial features of the raw EMEG signals and device layout.

The BrainTokenizer consists of a SEANet encoder [74], which extracts temporal representations from EMEG signals, and a novel Sensor Encoder, which encodes sensor physical information into sensor embeddings. The temporal representation and the sensor embedding are then fused via a specialised cross-attention block, followed by residual vector quantisation (RVQ) [91] for quantisation.

**SEANet Encoder.** A SEANet encoder [74] is used to extract efficient and compact representations from EMEG signals in the temporal dimension, which is a convolutional temporal encoder with multi-layer stacked residual 1-dimensional (-dim) convolutional block and strided convolutional layers. It encodes the raw data $\mathbf{X}$ to a temporal representation $\mathbf{Z}_{\text{time}} \in \mathbb{R}^{C \times W \times D}$, where $W$ represents for latent steps and $D$ for feature dimensions.

**Sensor Encoder.** To handle input from different EMEG devices, a novel Sensor Encoder is proposed to support learnable sensor position encoding, which hierarchically fuses physical priors with learned

representations. As illustrated in Fig. 1, the Sensor Encoder contains a position embedding layer to encode the 3-dim Cartesian coordinates of sensor positions and sensor orientations $\mathbf{L} \in \mathbb{R}^{C \times 6}$, and a type embedding layer to encode sensor types $\mathbf{S} = \{s_1, s_2, \ldots, s_c | s_i \in \{0, 1, 2\}\}$. The sensor physical information $\mathbf{L}$ and $\mathbf{S}$ are then integrated to produce sensor embedding $\mathbf{V} \in \mathbb{R}^{C \times D}$.

**Channel Compression.** The temporal representation $\mathbf{Z}_{\text{time}}$ and the sensor embedding $\mathbf{V}$ are then fed into a specialised cross-attention block, which performs channel compression to unify signals with varying numbers of channels into a fixed number of latent source variable. The Value of the cross-attention block is set to the temporal representation, Key is the sum of temporal representation and sensor embedding, and a set of learnable Query is used to adaptively aggregate information from temporal and spatial representations, producing intermediate latent features $\mathbf{Z}_{\text{src}} \in \mathbb{R}^{C' \times W \times D}$, where $C'$ is the number of latent source variables determined as a hyperparameter. The cross attention operation is performed independently for each window, ensuring the consistency of channel compression between different temporal windows.

**Quantisation.** The latent features $\mathbf{Z}_{\text{src}}$ are discretised by a 4-layer RVQ module to discrete tokens $\mathbf{Q} \in \mathbb{R}^{C' \times W \times 4}$, which is used to train the BrainOmni model in Stage 2.

**Reconstructor.** In the reconstructor, the discrete tokens are first decoded into $\hat{\mathbf{Z}}_{\text{src}}$ by the RVQ decoder. Another cross-attention block is developed to convert the quantised latent source variables to continuous channel-wise temporal representations $\hat{\mathbf{Z}}_{\text{time}}$, followed by a SEANet decoder to reconstruct $\hat{\mathbf{X}}$, which is an inverted mirror of the encoder using transposed convolutions. It is important to note that the BrainTokenizer and the reconstructor correspond to the backward and forward solution, respectively, in traditional EEG/MEG source current activity estimation; a detailed explanation can be found in Appendix B.

**Training the BrainTokenizer.** The BrainTokenizer is trained following an autoencoder scheme where 25% of the input channels are randomly dropped and the reconstructor is required to reconstruct all original channels from discrete tokens. The overall training loss combines a multi-level reconstruction loss with RVQ commitment losses, enabling joint optimisation of the network and the codebooks. The multi-level reconstruction loss includes **(i)** a time-domain loss between the original waveform $\mathbf{X}$ and the reconstructed waveform $\hat{\mathbf{X}}$:

$$\mathcal{L}_{\text{time}} = ||\mathbf{X} - \hat{\mathbf{X}}||, \tag{1}$$

where $|| \cdot ||$ denotes L1 distance; **(ii)** a frequency-domain loss between the original and reconstructed amplitude spectrum $\mathbf{A}$ and phase spectrum $\mathbf{\Phi}$:

$$\mathcal{L}_{\text{freq}} = ||\mathbf{A} - \hat{\mathbf{A}}|| + ||\mathbf{\Phi} - \hat{\mathbf{\Phi}}||; \tag{2}$$

**(iii)** an auxiliary loss based on Pearson correlation coefficient (PCC) to regularise waveform trend consistency:

$$\mathcal{L}_{\text{pcc}} = e^{-\text{PCC}(\mathbf{X}, \hat{\mathbf{X}})}. \tag{3}$$

The codebooks of RVQ is updated using exponential moving average (EMA), with rotation trick employed as the gradient estimator. Donate $\mathbf{z}_i$ and $\mathbf{z}_{q_i}$ as the residual and nearest entry of $i^{\text{th}}$ layer in the codebook, respectively, the RVQ commitment loss is defined as:

$$\mathcal{L}_{\text{rvq}} = \sum_{i=1}^{N_q} ||\mathbf{z}_i - \mathbf{z}_{q_i}||^2, \tag{4}$$

where $N_q$ is the depth of codebooks. The BrainTokenizer is trained to optimise the following loss:

$$\mathcal{L}_{\text{token}} = \mathcal{L}_{\text{time}} + \mathcal{L}_{\text{freq}} + \mathcal{L}_{\text{pcc}} + \mathcal{L}_{\text{rvq}}. \tag{5}$$

## 2.3 BrainOmni

Built on BrainTokenizer in Stage 1, the BrainOmni model is trained with a masked token prediction framework, to jointly model spatial and temporal information, thus learning coherent spatiotemporal representations of the brain activity. The training framework of BrainOmni is illustrated in Fig. 2.

The BrainTokenizer encodes continuous EMEG recordings into a sequence of tokens with shape $(C', T, N_q)$, where $C'$ is the number of the channels after compression, $T$ is the sequence length, and

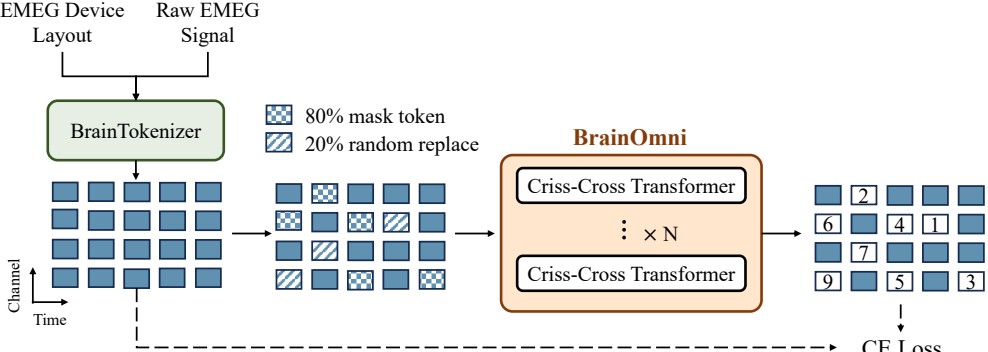

Figure 2: Illustration of the training framework of BrainOmni.

$N_q$ is the codebook depth, which is set to 4 in our implementation. Tokens are randomly masked at a predefined ratio and subsequently projected through an embedding layer into representations of shape $(C', T, D)$. The embeddings are then processed by multiple layers of the Criss-Cross Transformer [78] to jointly model the spatial and temporal dependencies. Finally, the output representations from Transformer blocks are utilised to predict the masked tokens.

**Criss-Cross Transformer.** The BrainOmni model consists of multiple Criss-Cross Transformer blocks [78]. Given the multi-channel time-series nature of EMEG data, joint modelling of spatial and temporal contexts is crucial. The Criss-Cross Transformer divides the input into two halves along the feature dimension: one half is used for spatial attention computation, and the other half for temporal attention. These two halves are then concatenated and passed through a feedforward layer. Additionally, rotary position embedding (RoPE) [72] is applied specifically to temporal attention to encode positional information along the time dimension.

**Training the BrainOmni Model.** During BrainOmni training, 50% of the positions in the $(C', T)$ token grid are randomly masked. For each masked position, all four RVQ layers are masked and simultaneously predicted in a non-autoregressive manner.

To prevent over-reliance on special mask tokens, 80% of the masked positions are replaced with dedicated mask tokens, while the remaining 20% are substituted with randomly sampled tokens from the sequence. The model training loss is:

$$\mathcal{L}_{\text{model}} = \frac{1}{M} \sum\nolimits_{i=1}^{M} \sum\nolimits_{j=1}^{N_q} \mathcal{L}_{\text{ce}}(q_{ij}, y_{ij}), \tag{6}$$

where $q_{ij}$ is the ground-truth codebook index at $j^{\text{th}}$ RVQ layer of token $i$, $y_{ij}$ is the corresponding model prediction, $M$ denotes the number of masked tokens, and $\mathcal{L}_{\text{ce}}(\cdot, \cdot)$ is the cross-entropy loss.

## 3 Experimental Setup

### 3.1 Pretraining

**Pretraining Datasets.** 1997 hours of EEG data and 656 hours of MEG data collected from open-source datasets are used for pretraining (see Appendix H for details), including EEG devices with channel numbers ranging from 19 to 128 and MEG devices with channel numbers ranging from 157 to 306. Among them, 85% of the data is used for training, 10% of the data is used for validation, and 5% of the data is used for testing. Additionally, one EEG dataset and one MEG dataset, which were both collected with unique device systems different from those in the training data, were excluded from the training data to evaluate the model's cross-device generalisation ability.

**Data Preprocessing.** Minimal and standardised preprocessing was applied to maximise data utilisation. A 0.1Hz-96 Hz bandpass filter and 50/60 notch filters were first applied to remove slow drifts and power-line noise, and MEG recordings with HPI coils were additionally filtered in their

Table 1: Overview of the downstream datasets. Duration in seconds.

| Modality | Task | Dataset | # Subject | # Channel | Duration | # Segment | # Class |
|---|---|---|---|---|---|---|---|
| EEG | Alzheimer's Disease | AD65 [43] | 65 | 19 | 10.0 | 5349 | 2 |
| | Depression | MDD [44] | 35 | 20 | 10.0 | 7302 | 2 |
| | Parkinson's Disease | PD31 [56] | 31 | 32 | 10.0 | 882 | 2 |
| | Abnormal | TUAB [47] | 2328 | 21 | 10.0 | 408853 | 2 |
| | Event | TUEV [47] | 294 | 21 | 5.0 | 112237 | 6 |
| | Emotion | FACED [9] | 123 | 30 | 10.0 | 10332 | 3 |
| | Motor Imagery | WBCIC_SHU [83] | 51 | 58 | 4.0 | 30591 | 2 |
| | Motor Imagery | PhysioNet-MI [58] | 109 | 64 | 4.0 | 9837 | 4 |
| MEG | Autism Spectrum Disorder | ASD74 [19] | 74 | 306 | 10.0 | 12320 | 2 |
| | Depression | MEG-MMI [40] | 51 | 269 | 30.0 | 1770 | 2 |
| EMEG | Motor Response | SomatoMotor [37] | 5 | 372 | 2.0 | 1208 | 2 |

HPI frequency bands. All signals were then resampled to 256 Hz. Bad channels were identified via a power-spectral-density-based detection algorithm (see Appendix E for details) and subsequently interpolated. To address reference inconsistencies and wide amplitude variations, reference values are first subtracted across all sensors of each type within each sample. Then, each channel is normalised to zero mean and unit variance at the sample level. Sensor coordinates and orientations are obtained from either the dataset-provided positions or the device's standard montage.

## 3.2 Evaluation

**Downstream Datasets.** Eleven different datasets for nine separate tasks are collected to evaluate the performance of BrainOmni. Overview of datasets, including modality, tasks, number of subjects, channels, segments, categories, and duration of each segment, are listed in Table 1, and details are listed in Appendix I. To prevent information leakage, all datasets used for downstream testing were excluded from the pretraining process. Consistent with pretraining, bandpass filtering, notch filtering, resampling and normalisation were applied for downstream dataset preprocessing.

**Baselines and Downstream Settings.** The proposed method is compared to four specialised EEG models: CNN-Transformer [52], ContraWR [85] SPaRCNet [35] and ST-Transformer [70], and a specialised MEG model: FAMED [31]. The specialised models were implemented without pretraining. Two additional EEG foundation models are included for EEG data: LaBraM [33] and CBraMod [78]. Details of these baselines can be found in Appendix J.

To ensure fair comparison, BrainOmni is compared to all baselines using the same preprocessing procedures, training pipeline and downstream evaluation strategy. The only exception is that in the preprocessing of LaBraM and CBraMod, we followed the same setting of sampling rate and bandpass filter as their preprocessing. All experiments were conducted under a 5-fold cross-validation setup, where each split allocated three folds for training, one for validation, and one for testing. Each configuration was run under two random seeds. To evaluate the model's generalisation across subjects, a strict cross-subject split strategy was applied to all datasets where subjects in the training set do not appear in the validation or test sets. Given class imbalances in some datasets, balanced accuracy (BACC) was chosen as the primary evaluation metric. The mean and standard deviation of all 10 runs were reported. Models were trained for 30 epochs, and the model checkpoint that achieved the highest validation-set BACC was selected for testing. More details of downstream training can be found in Appendix D.

## 4 Results

### 4.1 Downstream Evaluation

BrainOmni was evaluated on downstream EEG, MEG, and EMEG tasks. Results on EEG datasets are shown in Table 2. Results on MEG and EMEG datasets are listed in Table 3. It can be observed that BrainOmni achieves the highest performance on all tasks, with a close second-best place on the PhysioNet-MI dataset. Specifically, on EEG data, BrainOmni_base's balanced accuracy on the AD65 and PD31 datasets is about 10% higher than the best pretrained baseline LaBraM. On the

Table 2: Baseline comparison on the eight EEG datasets. Balanced accuracy is reported with mean $\pm$ standard deviation. "PT" stands for whether the model involves pretraining. The best results are shown in bold, and the second-best results are underlined.

| | # Param | PT | AD65 | MDD | PD31 | TUAB |
|---|---|---|---|---|---|---|
| CNN-Transformer [52] | - | ✗ | 0.695 ± 0.118 | 0.846 ± 0.096 | 0.536 ± 0.090 | 0.795 ± 0.024 |
| ContraWR [85] | - | ✗ | 0.660 ± 0.104 | 0.863 ± 0.068 | 0.521 ± 0.069 | 0.800 ± 0.012 |
| SPaRCNet [35] | - | ✗ | 0.582 ± 0.085 | 0.809 ± 0.051 | 0.534 ± 0.081 | 0.779 ± 0.017 |
| ST-Transformer [70] | - | ✗ | 0.604 ± 0.080 | 0.835 ± 0.040 | 0.530 ± 0.081 | 0.793 ± 0.008 |
| LaBraM [33] | 5.8M | ✓ | 0.711 ± 0.060 | 0.880 ± 0.037 | 0.659 ± 0.188 | 0.816 ± 0.006 |
| CBraMod [78] | 4.9M | ✓ | 0.681 ± 0.040 | 0.871 ± 0.048 | 0.584 ± 0.127 | 0.808 ± 0.007 |
| BrainOmni_tiny | 8.4M | ✓ | 0.795 ± 0.030 | **0.886 ± 0.043** | 0.736 ± 0.116 | **0.819 ± 0.004** |
| BrainOmni_base | 33M | ✓ | **0.828 ± 0.030** | 0.877 ± 0.052 | **0.748 ± 0.139** | 0.819 ± 0.005 |

| | # Param | PT | TUEV | FACED | WBCIC_SHU | PhysioNet-MI |
|---|---|---|---|---|---|---|
| CNN-Transformer [52] | - | ✗ | 0.331 ± 0.047 | 0.357 ± 0.011 | 0.666 ± 0.008 | 0.337 ± 0.019 |
| ContraWR [85] | - | ✗ | 0.371 ± 0.030 | 0.347 ± 0.011 | 0.579 ± 0.031 | 0.385 ± 0.026 |
| SPaRCNet [35] | - | ✗ | 0.362 ± 0.036 | 0.377 ± 0.012 | 0.748 ± 0.018 | 0.456 ± 0.016 |
| ST-Transformer [70] | - | ✗ | 0.392 ± 0.032 | 0.401 ± 0.008 | 0.749 ± 0.007 | 0.398 ± 0.015 |
| LaBraM [33] | 5.8M | ✓ | 0.588 ± 0.017 | 0.458 ± 0.014 | 0.831 ± 0.015 | 0.561 ± 0.015 |
| CBraMod [78] | 4.9M | ✓ | 0.525 ± 0.021 | 0.441 ± 0.014 | 0.822 ± 0.016 | **0.595 ± 0.015** |
| BrainOmni_tiny | 8.4M | ✓ | 0.603 ± 0.024 | 0.472 ± 0.018 | 0.825 ± 0.015 | 0.580 ± 0.019 |
| BrainOmni_base | 33M | ✓ | **0.622 ± 0.028** | **0.490 ± 0.013** | **0.832 ± 0.013** | 0.590 ± 0.022 |

Table 3: Baseline comparison on the ASD74 (MEG), MEG-MMI (MEG) and SomatoMotor (EMEG). Balanced accuracy are reported in mean $\pm$ standard deviation. "PT" stands for whether the model involves pretraining. The best results are shown in bold and second-best results are underlined.

| | # Param | PT | ASD74 | MEG-MMI | SomatoMotor |
|---|---|---|---|---|---|
| FAMED [31] | - | ✗ | 0.555 ± 0.038 | 0.527 ± 0.100 | 0.500 ± 0.016 |
| CNN-Transformer [52] | - | ✗ | 0.517 ± 0.034 | 0.549 ± 0.074 | 0.503 ± 0.014 |
| ContraWR [85] | - | ✗ | 0.502 ± 0.009 | 0.528 ± 0.035 | 0.491 ± 0.020 |
| SPaRCNet [35] | - | ✗ | 0.604 ± 0.053 | 0.581 ± 0.109 | 0.541 ± 0.027 |
| ST-Transformer [70] | - | ✗ | 0.583 ± 0.034 | 0.561 ± 0.054 | 0.661 ± 0.047 |
| BrainOmni_tiny | 8.4M | ✓ | 0.621 ± 0.048 | **0.610 ± 0.057** | **0.863 ± 0.128** |
| BrainOmni_base | 33M | ✓ | **0.651 ± 0.043** | 0.604 ± 0.061 | 0.832 ± 0.064 |

MEG ASD74 dataset, it exceeds the strongest baseline by about 5%, and on the EMEG SomatoMotor task, it outperforms the top baseline by 20%. The results demonstrate that BrainOmni consistently outperforms both specialised and other pretrained foundation models across diverse downstream tasks and modalities, including EEG, MEG, and EMEG data, exhibiting strong generalisation and versatility.

## 4.2 Cross Device Generalisation

To evaluate BrainTokenizer's generalisation to entirely unseen EEG and MEG device systems, we selected two datasets, PerceiveImagine [6] (EEG, *SynAmps2* system) and Gloups-MEG [76] (MEG, *NeuroImaging* system), whose recording setups were not included in pretraining data. We compare the zero-shot reconstruction losses of these unseen datasets against those of the known-device EEG and MEG test set whose training partitions have been used in training BrainTokenizer.

Results are shown in Table 4. For EEG, the zero-shot results on PerceiveImagine even consistently outperform our EEG test set across all metrics, highlighting strong generalisation to a new EEG device system. For MEG, although the reconstruction losses of BrainTokenizer on Gloups-MEG are slightly worse than those of our own MEG test set, the relatively strong PCC value of 0.695 still indicates the model's competitive generalisation capability for MEG devices. The slight performance degradation on MEG devices may be attributed to the relatively limited amount of MEG training data.

Table 4: Zero-shot reconstruction results of BrainTokenizer on unseen devices. "AMP" stands for amptitude MAE. PHASE stands for phase MAE. "↑" indicates the higher the better. "↓" indicates the lower the better.

| Modality | Data | Seen Device | MSE ↓ | MAE ↓ | AMP ↓ | PHASE ↓ | PCC ↑ |
|---|---|---|---|---|---|---|---|
| EEG | our EEG Test Set | ✓ | 0.404 | 0.449 | 0.142 | 1.506 | 0.748 |
| | PerceiveImagine [6] | ✗ | 0.343 | 0.415 | 0.137 | 1.457 | 0.802 |
| MEG | our MEG Test Set | ✓ | 0.473 | 0.522 | 0.220 | 1.469 | 0.711 |
| | Gloups-MEG [76] | ✗ | 0.567 | 0.572 | 0.206 | 1.517 | 0.695 |

Table 5: Comparision of BrainOmni models pretrained using EEG-only, MEG-only, and joint EMEG data. Tiny version is used for all models.

| Dataset | TUAB | TUEV | ASD74 | SomatoMotor | | |
|---|---|---|---|---|---|---|
| Modality | EEG | EEG | MEG | EEG | MEG | EMEG |
| EEG only | $0.811 \pm 0.006$ | $0.583 \pm 0.020$ | – | $0.766 \pm 0.070$ | – | – |
| MEG only | – | – | $0.553 \pm 0.062$ | – | $0.802 \pm 0.141$ | – |
| EMEG | $\mathbf{0.819 \pm 0.004}$ | $\mathbf{0.603 \pm 0.024}$ | $\mathbf{0.621 \pm 0.048}$ | $\mathbf{0.783 \pm 0.076}$ | $\mathbf{0.838 \pm 0.128}$ | $\mathbf{0.863 \pm 0.128}$ |

## 4.3 EMEG Joint Pretraining

To investigate the effect of joint EEG and MEG training on model performance, two BrainOmni variants were trained using only EEG or only MEG data. The variants are compared to the BrainOmni model on EEG datasets (TUAB, TUEV), MEG dataset (ASD74), and multimodal dataset (Somato-Motor). For the SomatoMotor dataset, downstream experiments were conducted with EEG-only, MEG-only, and combined EMEG inputs separately. As shown in Table 5, joint EMEG pretraining consistently outperforms single-modality pretraining across all datasets. Notably, for the MEG ASD74 dataset, the jointly pretrained EMEG model shows a 12% relative BACC improvement compared with the MEG-only model. The results highlight the benefit of multimodal training, particularly for the MEG modality with less accessible pretraining data. Additionally, comparing the EEG, MEG, and EMEG inputs processed by the EMEG model on SomatoMotor, the combined EMEG input achieves better performance than both EEG and MEG inputs, underscoring the effectiveness of jointly leveraging EEG and MEG data in downstream tasks.

# 5 Analysis

## 5.1 Effectiveness of Sensor Encoder

An ablation study is conducted to assess the effectiveness of the proposed Sensor Encoder in modelling the spatial characteristics of sensors. Apart from directly removing the sensor embedding, we also develop a pure temporal version of BrainTokenizer where multi-channel EMEG data is treated as uncorrelated single-channel data. As shown in Table 6, the standard BrainOmni consistently outperforms the pure temporal model. And the exclusion of sensor embedding significantly undermines the downstream performance, especially on challenging MEG and EMEG datasets, which demonstrates the effectiveness of the proposed Sensor Encoder.

## 5.2 Impact of Loss Items in BrainTokenizer

The BrainTokenizer training loss contains four components: time-domain loss, frequency-domain loss, Pearson correlation loss, and RVQ commitment loss. Given that the time-domain and commitment losses are essential for training, we conducted an ablation study for how frequency loss and PCC loss

Table 6: Ablation study of sensor embedding (SE). Tiny version used. Balanced accuracy reported.

| Model | TUAB | TUEV | AD65 | ASD74 | SomatoMotor |
|---|---|---|---|---|---|
| BrainOmni | $\mathbf{0.819 \pm 0.004}$ | $0.603 \pm 0.024$ | $\mathbf{0.795 \pm 0.030}$ | $\mathbf{0.621 \pm 0.048}$ | $\mathbf{0.863 \pm 0.128}$ |
| –w/o SE | $0.784 \pm 0.005$ | $\mathbf{0.607 \pm 0.016}$ | $0.730 \pm 0.064$ | $0.568 \pm 0.070$ | $0.744 \pm 0.076$ |
| Pure temporal | $0.788 \pm 0.003$ | $0.526 \pm 0.016$ | $0.763 \pm 0.047$ | $0.571 \pm 0.053$ | $0.766 \pm 0.050$ |

Table 7: Reconstruction results of BrainTokenizer w or w/o freq loss and PCC loss. Total loss denotes the sum of all four loss items.

| Freq loss | PCC loss | MAE ↓ | AMP ↓ | PHASE ↓ | PCC ↑ | Commitment ↓ | Total Loss ↓ |
|-----------|----------|-------|-------|---------|-------|--------------|--------------|
|           |          | 0.799 | 0.256 | 2.154   | 0.002 | 5.01e-7      | 4.207        |
| ✓         |          | 0.471 | 0.160 | 1.583   | 0.716 | 1.64e-4      | 2.517        |
|           | ✓        | 0.462 | 0.167 | 1.863   | 0.751 | 1.83e-4      | 2.703        |
| ✓         | ✓        | 0.480 | 0.165 | 1.495   | 0.723 | 1.48e-4      | 2.626        |

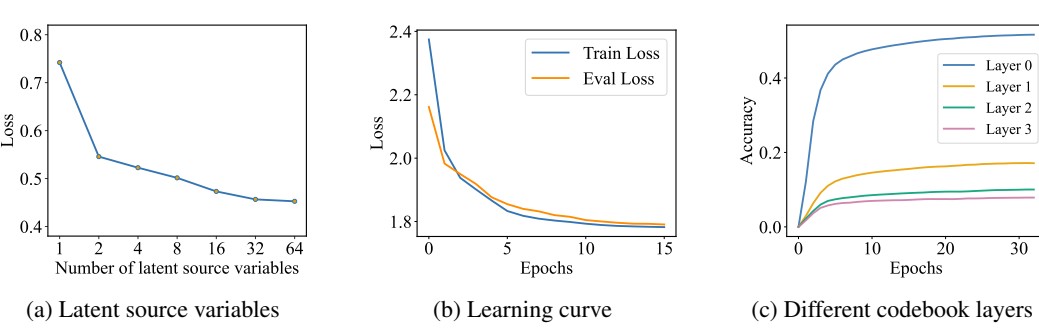

(a) Latent source variables     (b) Learning curve     (c) Different codebook layers

Figure 3: (a) Trend of L1 loss over the number of latent source variables. (b) The training loss and validation loss curves during the training phase of BrainTokenizer. (c) The accuracy curves for parallel mask prediction on the labels of each codebook layer during the training phase of BrainOmni on the training set.

influence the reconstruction performance. Results are shown in Table 7. With only time-domain and commitment loss, the decoder outputs collapsed into near-constant signals and got poor performance. Introducing the frequency-domain loss greatly improved spectral detail (lower MAE/AMP/PHASE), but led to spurious high-frequency impulses. The PCC term mitigates these impulses by aligning overall waveform trends, yet on its own provides weaker spectral reconstruction. All four components were included in the BrainTokenizer training, suppressing the impulse while preserving as much frequency-domain detail as possible.

### 5.3 Number of Latent Source Variables

To investigate the effect of the number of latent source activities, we trained variants of BrainTokenizer with different numbers of latent source variables. Results are shown in Fig. 3a. As the number of source activities increases, the EMEG signal reconstruction loss gradually decreases. When the number of source variables reaches beyond 16, the trend of loss reduction weakened with further increases in the number of latent source activities. To balance the amount of information retained and computational efficiency, we chose 16 as the number of latent source variables.

### 5.4 Pretraining Stability and Codebook Contribution

To analyse the training stability when incorporating diverse devices and signals into pretraining, we plot the training curves in Fig. 3b. Overall, BrainTokenizer converges relatively quickly, and the validation loss remains consistent with the training loss. This demonstrates that the proposed BrainTokenizer training framework handle different devices and signal types well. Fig. 3c plots the mask prediction performance for each RVQ layer. It can be seen that the first codebook aggregates richer semantic information and achieves higher accuracy in mask prediction, while the prediction accuracy decreases progressively for the subsequent lower-level codebooks.

### 5.5 Visualisation of Reconstruction Results

Fig. 4 illustrates the reconstruction performance of BrainTokenizer. We compare the waveforms and topographic maps before and after reconstruction to show the model's ability to capture the spatiotemporal electromagnetic fields of EEG and MEG in both temporal and spatial domains. The results show that the model effectively preserves the main trends and finer details of the waveforms,

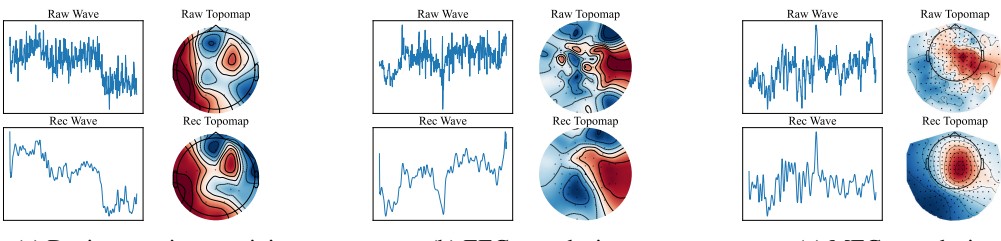

(a) Device seen in pretraining.     (b) EEG new device.     (c) MEG new device.

Figure 4: The waveforms and topographies of the reconstructed and original EEG signals: (a) the standard *10-20* system that was seen during the pre-training phase; (b) the *Synnaps* system not seen during the pretraining phase; (c) the *NeuroImaging* system not seen during the pretraining phase.

while smoothing out high-frequency noise spikes. In terms of spatial representation, the reconstructed topographic maps maintain the original activation patterns and structural integrity. Notably, the model also achieves strong reconstruction performance on previously unseen devices and samples, demonstrating robust generalisation across different recording setups.

## 6 Conclusion

In this paper, we introduce BrainOmni, the first brain foundation model that generalises across heterogeneous EEG and MEG signals. To model signals from different devices and modalities, we propose BrainTokenizer, which infers spatiotemporal patterns of brain activity from the observed EMEG signals and generates quantized discrete tokens. To address the inherent heterogeneity arising from various recording devices, we develop a flexible Sensor Encoder that leverages physical sensor properties. Through large-scale self-supervised joint pretraining on EEG and MEG data, BrainOmni significantly exceeds the performance of existing foundation models and state-of-the-art task-specific baselines across various downstream tasks, and demonstrates effective generalisation capability to unseen devices and modalities. Furthermore, extensive ablation analysis highlights the consistent benefits of joint EMEG pretraining, underscoring the advantage of unified modelling strategies. We believe BrainOmni represents an important step towards building versatile and scalable foundation models for neural recordings, opening new pathways for unified brain signal representation learning.

## 7 Acknowledgements

This work was supported by the New Generation Artificial Intelligence – National Science and Technology Major Project of China (Grant No. 2025ZD0121803) and the National Natural Science Foundation of China (Grant No. 62476151).

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

## A  Broader Impacts and Limitations

As the first foundation model enabling unified analysis across EEG and MEG modalities, BrainOmni represents a crucial step toward generalizable and scalable neural representation learning, offering substantial application potential in neuroscience research, health monitoring, and clinical diagnosis. Additionally, the model's enhanced generalisation on heterogeneous devices could reduce the performance degradation typically associated with device and dataset variability. While it may not always outperform domain-specific models tailored to one dataset or one task, it offers broader coverage and generalisability across datasets, tasks, and modalities – a key advantage for scalable and reusable neuro-AI systems.

Here we analyse the limitations of the work. Firstly, the scale of EEG and MEG data used for pretraining is still relatively limited. In particular, due to the limited availability of publicly accessible MEG datasets, only 656 hours of MEG data were collected for the pretraining phase. Secondly, the downstream evaluation on MEG and EMEG modalities is constrained by both the scarcity of suitable datasets and the current lack of standardised test paradigms. In future work, we plan to incorporate a larger and more diverse collection of datasets, especially across the MEG and EMEG modalities, to strengthen model performance and validate generalisation. Furthermore, we will extend our framework to include additional modalities of neural signals beyond EEG and MEG to develop a more comprehensive and unified representation learning approach for neural activity recordings.

## B  Neuroscience Background

### B.1  Source Current Estimation

Source current estimation seeks to infer the spatiotemporal distribution of neural currents within the brain from non-invasive scalp measurements. It is an "ill-posed" inverse problem: it is impossible to unambiguously determine the three-dimensional source current distribution inside the brain that gave rise to those measurements even perfectly record full electric and magnetic field distribution around the head [29]. The distributed source model [12, 50] was developed to solve this problem, using a large number of dipoles or monopoles distributed within the brain volume or the cortex, making the problem linear. A commonly used and well-tested method is minimun norm estimation (MNE), which obtains the maximum a posteriori probability estimate of source activity by constructing a linear inversion operator $\mathbf{W}$ under regularized conditions [25]. This method has been commonly used in multiple neuroscience domains, including but not limited to audio decoding [73], emotion analysis [53], and epilepsy diagnosis [3, 18].

Broadly speaking, MNE involves two main stages: first, computing the forward solution (also known as the leadfield matrix); and second, estimating the inverse operator (or backward solution) that maps sensor data back to source space.

**The Forward Solution**  The forward problem in EEG and MEG refers to computing the scalp electric potentials or magnetic fields that would be measured given a known distribution of source currents within the brain. Under the quasi-static approximation of Maxwell's equations, the relationship between sources and measurements is linear and can be expressed as:

$$\mathbf{y}(t) = \mathbf{L}\mathbf{j}(t) + \mathbf{n}(t) \tag{7}$$

where $\mathbf{y}(t) \in \mathbb{R}^M$ is the vector of sensor measurements at time $t$, $\mathbf{j}(t) \in \mathbb{R}^N$ is the source current distribution, $\mathbf{L} \in \mathbb{R}^{M \times N}$ is the leadfield matrix, and $\mathbf{n}(t)$ is measurement noise [25, 5].

**Estimating the leadfield Matrix from the Head Geometry and Conductivity**  The leadfield matrix $\mathbf{L}$ encapsulates how each source location contributes to the sensor array and is calculated using a head model. This model is constructed from anatomical MRI, using numerical methods such as the boundary element method (BEM) or finite element method (FEM), and incorporates realistic tissue conductivities.

The human head is typically modelled as a piecewise homogeneous volume conductor with compartments representing scalp, skull, cerebrospinal fluid (CSF), and brain. Each region is assigned an

isotropic or anisotropic conductivity value $\sigma_i$, and the electrical potential $\phi$ in the volume conductor is governed by the Poisson equation:

$$\nabla \cdot (\sigma(\mathbf{r})\nabla\phi(\mathbf{r})) = \nabla \cdot \mathbf{J}_p(\mathbf{r}) \tag{8}$$

where $\sigma(\mathbf{r})$ is the spatially varying conductivity, and $\mathbf{J}_p(\mathbf{r})$ is the primary current density, confined to the gray matter (cortex). For EEG, the scalp potential $V$ at sensor location $\mathbf{r}_m$ is obtained by solving this equation using numerical methods like the boundary element method (BEM) or finite element method (FEM):

$$V(\mathbf{r}_m) = \int_\Omega G(\mathbf{r}_m, \mathbf{r}')\nabla \cdot \mathbf{J}_p(\mathbf{r}')\, d\mathbf{r}' \tag{9}$$

Here, $G(\mathbf{r}_m, \mathbf{r}')$ is the Green's function representing the head's geometry and conductivity structure, while $\Omega$ is the brain/source volume. The leadfield matrix $\mathbf{L}$ used in the forward model is computed by solving this equation for each source element and each sensor.

In the case of MEG, the magnetic field $\mathbf{B}$ generated by a primary current $\mathbf{J}_p$ is computed using the quasi-static Biot-Savart law:

$$\mathbf{B}(\mathbf{r}_m) = \frac{\mu_0}{4\pi} \int_\Omega \frac{\mathbf{J}_p(\mathbf{r}') \times (\mathbf{r}_m - \mathbf{r}')}{|\mathbf{r}_m - \mathbf{r}'|^3}\, d\mathbf{r}' \tag{10}$$

Since MEG is insensitive to radial currents and largely unaffected by conductivity discontinuities (like the low-conductivity skull), it is more robust to inaccuracies in the conductivity model [5]. In contrast, EEG is highly sensitive to these properties, especially the skull's low conductivity ($\sim 0.0042$ S/m compared to the brain's $\sim 0.33$ S/m) [77].

To construct realistic forward models, anatomical MRI data is segmented to extract surfaces or volumes of different tissue types. BEM solves the boundary integrals over these surfaces, assuming piecewise constant conductivities, while FEM can accommodate more complex, anisotropic conductivity distributions within each region [48].

**The Backward Solution and Linear Inverse Operator** The inverse problem – estimating the sources $\mathbf{j}(t)$ from the measurements $\mathbf{y}(t)$ – is ill-posed and requires regularization. Minimum-norm estimation solves this by seeking the source configuration with the smallest $\ell_2$-norm that still explains the data:

$$\hat{\mathbf{j}}(t) = \arg\min_{\mathbf{j}} \|\mathbf{y}(t) - \mathbf{L}\mathbf{j}(t)\|_2^2 + \lambda^2 \|\mathbf{j}(t)\|_2^2 \tag{11}$$

where $\lambda$ is a regularization parameter controlling the trade-off between data fidelity and source power [25, 28].

This yields a closed-form solution:

$$\hat{\mathbf{j}}(t) = \mathbf{W}\mathbf{y}(t), \quad \text{where} \quad \mathbf{W} = \mathbf{L}^\top \left(\mathbf{L}\mathbf{L}^\top + \lambda^2 \mathbf{C}_n\right)^{-1} \tag{12}$$

Here, $\mathbf{C}_n$ is the noise covariance matrix. When whitening is applied, we define:

$$\mathbf{L}_w = \mathbf{C}_n^{-1/2}\mathbf{L}, \quad \mathbf{y}_w(t) = \mathbf{C}_n^{-1/2}\mathbf{y}(t) \tag{13}$$

and compute the inverse operator as:

$$\mathbf{W} = \mathbf{G}\mathbf{L}_w^\top \left(\mathbf{L}_w\mathbf{L}_w^\top + \lambda^2 \mathbf{I}\right)^{-1} \tag{14}$$

where $\mathbf{G}$ is a source covariance matrix, often the identity or a depth-weighted diagonal matrix to compensate for depth bias [38].

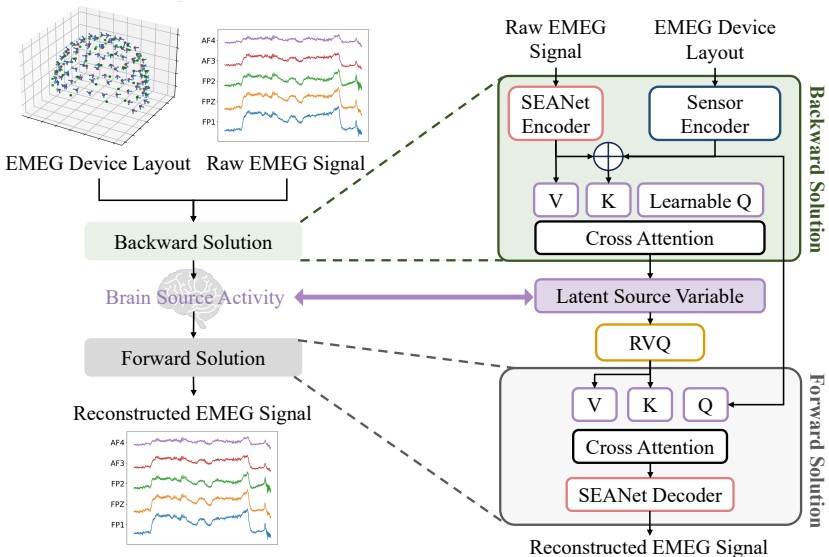

Figure 5: Correlation between source current estimation and the proposed BrainTokenizer. The BrainTokenizer (excluding RVQ) can be viewed as the backward solution in source current estimation, and the reconstructor can be viewed as the forward solution.

## B.2 Model Explanation From the Perspective of Neuroscience

Our BrainTokenizer design closely parallels the classical source current estimation process, as illustrated in Fig. 5. As introduced above, the backward solution in source current estimation infer brain source activity from externally measured signals and sensor layout information, and forward solution reconstructs the external signals from the inferred brain source activity.

Inspired by this conceptual framework, our proposed BrainTokenizer closely mirrors these two steps. Specifically, the BrainTokenizer (excluding the RVQ quantisation step) corresponds directly to the backward solution stage, where it maps the raw EMEG signals and sensor device layout into latent source variables through a parametric module. Recall the classical backward solution,

$$\hat{\mathbf{j}}(t) = \mathbf{W}\mathbf{y}(t), \quad \text{where} \quad \mathbf{W} = \mathbf{L}^\top \left(\mathbf{L}\mathbf{L}^\top + \lambda^2 \mathbf{C}n\right)^{-1} \tag{15}$$

which relies on a fixed linear inverse operator $\mathbf{W}$ constructed from a leadfield matrix $\mathbf{L}$ and a noise covariance matrix $\mathbf{C}_n$. While in our architecture, we use the attention weights in cross-attention mechanisms to construct a time-varying parametric inverse operator $\mathbf{W}_\theta(\mathbf{Z}_{\text{time}}, \mathbf{L}, \mathbf{S})$, where the operator is related to sensor prosperity and the measured data since the different fields recorded outside may indicate that there is a change in the pattern of neuro activity, formulated as:

$$\mathbf{K} = \mathbf{Z}_{\text{time}} + \mathbf{V}(\mathbf{L}, \mathbf{S}) \tag{16}$$

$$\mathbf{W}_\theta(\mathbf{Z}_{\text{time}}, \mathbf{L}, \mathbf{S}) = \frac{\text{softmax}(\mathbf{Q}\mathbf{K}^{\text{T}})}{\sqrt{d_{\text{head}}}} \tag{17}$$

where $\mathbf{Q}$ is a set of learnable query embeddings that dynamically aggregate temporal and spatial information, $\mathbf{K}$ is the combination of temporal representation $\mathbf{Z}_{\text{time}}$ from SEANet encoder and sensor embeddings $\mathbf{V}(\mathbf{L}, \mathbf{S})$, and $d_{\text{head}}$ is the scaling factor determined by the attention head dimension. Different from traditional linear, fixed operators, these learned attention weights naturally serve as flexible and data-driven inverse operators, adaptively integrating spatiotemporal sensor properties and temporal signal dynamics to estimate latent variables. Appendix G provides a visualisation for attention weights of cross attention.

Following this analogy, the reconstructor corresponds to the forward solution, reconstructing the original observed signals from the inferred latent sources. Similar to the backward process, the reconstructor also uses a cross-attention mechanism, now employing latent source variables as keys and sensor embeddings as values. Through the decoder module (SEANet decoder), these features are further translated back into reconstructed EMEG signals.

## C Related Work on Brain Foundation Models

A foundation model is a deep learning model pretrained on large-scale datasets, generally approached by self-supervised learning (SSL), to learn signal representations from abundant unlabelled data and reduce dependence on labelled samples. This paradigm has achieved notable success in computer vision [54, 92], natural language processing [17, 55, 49] and speech [4, 32, 10]. Brain signals, especially EEG and MEG, show low signal-to-noise ratio, high dimensionality and individual variability, making large-scale pretraining for robust representation and general brain model important. In EEG domain, BENDR [36], inspired by Wav2Vec 2.0 model, employs contrastive learning method to learn from massive anouts of EEG data. Since then, more model was proposed for more generalized and stronger representation. BIOT [84] enable cross-data learning with mismatched channels, variable lengths, and missing values by tokenizing different biosignals into unified "sentences" structure. MMM [89] adopts a topology-agnostic scheme and implements multi-dimensional position encoding, multi-level channel hierarchies and multi-stage pretraining. LaBraM [33] segments EEG signals into channel patches, discretizes them via vector quantization before training with masked EEG modeling to capture high-level semantics. CbraMod [78] introduces a Criss-Cross Transformer to fully leverage EEG's spatiotemporal characteristics. Beyond EEG-only approaches, multimodal brain foundation models have also been developed before. For instance, BrainWave [90] and PopT [8] jointly train on EEG and intracranial EEG (iEEG) data. However, despite MEG's superior spatiotemporal resolution and its frequent combined use with EEG in neuroscience, foundation model research for MEG and joint EMEG remains largely unexplored.

## D Implementation Details

We trained BrainTokenizer using 2-second segments. For training BrainOmni, we inputted 30-second data segments to allow the model to capture longer temporal dependencies. During the segmented tokenization process, we set the overlap ratio between windows to 25% to incorporate partial contextual information. The training was conducted on 16 A100 GPUs, using the AdamW optimizer and a warmup-cosine-decay learning rate scheduler with a warmup proportion of 10%. BrainTokenizer was trained for 16 epochs with a total batch size of 512 per update step and a maximum learning rate of 2e-4. BrainOmni was trained for 32 epochs with a total batch size of 256 per update step and a maximum learning rate of 4e-4. The BrainTokenizer training took approximately 11 hours, and BrainOmni required about 14 hours for the tiny model and 18 hours for the base model.

For downstream evaluation, all models follow a unified training pipeline. The output embeddings from each model are first average pooled along the temporal dimension, then flattened across remaining dimensions to serve as feature, which are subsequently fed into a two-layer MLP for classification. Taking BrainOmni as an example, for a batch of samples, the model outputs a embedding with shape $(B, C', T, D)$, where $B$ represents for batch size, $C'$ the number of compressed channels, $T$ temporal sequence length, and $D$ embedding dimension. The embeddings is firstly average pooled over $T$, and then flattened to $(B, C' * D)$, before fed into the MLP classifier.

The evaluation was conducted in a cross-subject five-fold cross-validation paradigm. For datasets without an official split (*i.e.*, datasets except TUAB and TUEV), all subjects were divided into five subsets, and five experiments were conducted in rotation. In each experiment, three folds were used for training, one for validation, and one for testing, ensuring that every subset served as the test set once, thereby providing a more comprehensive assessment of the model's performance. For TUAB and TUEV which provided an official train/eval split, we kept the eval set for all testing, and split the official training subjects equally into five folds, and in each runs use four folds for training and remaining fold for validation. All experiments were run under two random seeds (42 and 3407) and seed 42 is used for data splitting. Experiments were conducted with learning rates of $[3 \times 10^{-6}, 1 \times 10^{-5}, 3 \times 10^{-5}]$ for each model, and the learning rate yielding the highest result was selected to obtain the best performance of each model. All other hyperparameters remained consistent across experiments (specific values are provided in Table 12). On a single A100 GPU, the training throughput was approximately 60 samples/sec, while inference achieved roughly 90 samples/sec.

## E Preprocess Details

The power-spetral density-based bad-channel detection is implemented as Alg. 1.

**Algorithm 1:** Power-spectral-density-based bad-channel detection

---

**Input** : *raw_data*, *threshold* = 10
**Output** : *bad_channels*
**1. Compute per-channel PSD over the full recording:**
$\quad$ PSD $\leftarrow$ COMPUTEPSD(*raw_data*).*data*
**2. Stabilise and log-transform:**
$\quad$ $L \leftarrow \log(\text{PSD} + 10^{-16})$
**3. Compute pairwise distances between channel spectra:**
$\quad$ **for** $i \leftarrow 1$ **to** $C$ **do**
$\quad\quad$ **for** $j \leftarrow 1$ **to** $C$ **do**
$\quad\quad\quad$ $\text{dist}[i,j] \leftarrow L[i] - L[j]$
$\quad\quad$ **end**
$\quad$ **end**
**4. Mean distance per channel:**
$\quad$ $m[i] \leftarrow \text{mean}_j\big(\text{dist}[i,j]\big)$ for $i = 1, \ldots, C$
**5. Identify outliers via IQR:**
$\quad$ $Q_1 \leftarrow \text{percentile}_{25}(m), \quad Q_3 \leftarrow \text{percentile}_{75}(m)$
$\quad$ $\text{IQR} \leftarrow Q_3 - Q_1$
$\quad$ upper $\leftarrow Q_3 + \textit{threshold} \cdot \text{IQR}, \quad$ lower $\leftarrow Q_1 - \textit{threshold} \cdot \text{IQR}$
$\quad$ *bad_channels* $\leftarrow \{\, \text{ch}_i \mid m[i] > \text{upper} \ \vee \ m[i] < \text{lower} \,\}$

---

To address the reference inconsistency caused by different recording setups, we first compute a global average reference by taking the mean across all channels, and then subtract this channel-average signal from each channel waveform, effectively aligning all channels to the same virtual reference. This protocol is applied regardless of whether a reference channel is unavailable or the signals have already been referenced, and it is performed both at the recording level and separately for each signal type.

# F   Ablation Study

## F.1   Latent Source Activity Estimation

As mentioned in the Introduction, modelling at the electrode level can encounter issues such as low information density and redundant information in adjacent channels. The latent source activity modelling process involves compressing the sequence length, which, while condensing semantic features, may also introduce some loss of detail. To further demonstrate the advantages of modelling source activity compared to modelling at the electrode level, we trained a model that integrates spatial features through self-attention between electrodes, which would not involve loss of information. To be more specific, the BrainTokenizer of this model replaces the cross attention layer with self-attention layer. Channel random masking and additive noise are also applied before the self-attention layer in the encoder part of the feature input. In the decoder part, the masked channels are replaced with a mask token, and the waveform of the masked part is recovered by a self-attention layer. Results are show in Table 8.

We can see that even though models using latent source activity estimation may lose some information during the sequence length compression process, their performance on datasets such as TUAB (23), PD31 (32), AD65 (19), ASD74 (306) and SomatoMotor (372) still exceeds that of models built at the electrode level. The numbers in parentheses represent the number of channels in each dataset. Moreover, since the computational complexity of the Transformer model grows quadratically with sequence length, models using LSAE have significantly higher computational efficiency compared to those that do not use it. This demonstrates that the process of LSAE can eliminate redundant

Table 8: Balanced accuracy results of ablation study for latent source activity estimation (LSAE)

| | TUAB | PD31 | AD65 | ASD74 | SomatoMotor |
|---|---|---|---|---|---|
| tiny w/o LSAE | $0.813 \pm 0.003$ | $0.627 \pm 0.158$ | $0.768 \pm 0.056$ | $0.606 \pm 0.088$ | $0.801 \pm 0.078$ |
| tiny w/ LSAE | $\mathbf{0.819 \pm 0.004}$ | $\mathbf{0.736 \pm 0.116}$ | $\mathbf{0.795 \pm 0.030}$ | $\mathbf{0.621 \pm 0.048}$ | $\mathbf{0.863 \pm 0.128}$ |

Table 9: Downstream results with frozen backbone vs. full finetuning (BACC).

|  |  | TUAB | TUEV | AD65 |
|---|---|---|---|---|
| LaBraM | Freeze | $0.800 \pm 0.004$ | $0.432 \pm 0.024$ | $0.722 \pm 0.039$ |
|  | Full Finetune | $0.816 \pm 0.006$ | $0.588 \pm 0.017$ | $0.711 \pm 0.060$ |
| CBraMod | Freeze | $0.774 \pm 0.001$ | $0.345 \pm 0.014$ | $0.620 \pm 0.057$ |
|  | Full Finetune | $0.808 \pm 0.007$ | $0.525 \pm 0.021$ | $0.681 \pm 0.040$ |
| BrainOmni_tiny | Freeze | $0.800 \pm 0.002$ | $0.460 \pm 0.021$ | $0.774 \pm 0.048$ |
|  | Full Finetune | $0.819 \pm 0.004$ | $0.603 \pm 0.024$ | $0.795 \pm 0.030$ |
| BrainOmni_base | Freeze | $0.809 \pm 0.003$ | $0.480 \pm 0.015$ | $0.771 \pm 0.062$ |
|  | Full Finetune | $0.819 \pm 0.005$ | $0.622 \pm 0.028$ | $0.828 \pm 0.030$ |

information between channels while retaining core semantic features. It also illustrates the benefits of projecting all data from the electrode level into a unified feature space for modelling.

### F.2 Freeze Backbone for Downstream

In Table 9, we report the BACC metrics of BrainOmni and the baseline models under both full fine-tuning and frozen settings on TUAB, TUEV, and AD65 datasets. In TUAB and AD65, the performance of BrainOmni under the weight freezing scenario is comparable to or even higher than the metrics of the baseline models with full fine-tuning. This indicates that BrainOmni has learned the pattern features of EMEG signals with better generalisation capability through unsupervised training.

## G Attention Weights of Cross Attention in the BrainTokenizer

Recall the content in Appendix B.2, the cross-attention mechanism in BrainTokenizer is inspired by the inverse operator in the Backward Solution. The attention weights can actually be regarded as a parameterised linear inverse matrix, representing the degree of influence of each channel on each latent source activity. From Fig. 6, we can see that each source variable in our model has automatically learned a hierarchical structure. It is evident that each source variable focuses on extracting features from specific scalp regions. Moreover, in the multi-head attention mechanism, the attention of the first head is relatively concentrated, while the subsequent heads progressively expand the range of regions.

## H Pretraining Dataset Description

Below, we provide detailed descriptions of the datasets used for pretraining BrainTokenizer and BrainOmni.

- **MEG-MASC**[24]: The MEG-MASC dataset includes MEG recordings (208 channels, 1000 Hz) from 27 English speakers listening to 2 hours of naturalistic stories, collected using an axial-gradiometer KIT system.

- **MEG-Narrative-Dataset**[2]: The MEG-Narrative-Dataset includes MEG recordings (275 channels, 1200 Hz) from 3 English speakers listening to 10 hours of naturalistic stories, collected using an axial gradiometer CTF system.

- **OMEGA**[45]: The OMEGA dataset includes MEG recordings (306 channels, 1000 Hz) from 444 healthy subjects and 200 patient volunteers with Parkinson's disease, ADHD, and chronic pain in a resting state, collected using CTF whole-head MEG systems from VSM MedTech Inc.

- **CC700**[75] The CC700 dataset is a subset of the Cam-CAN dataset, including MEG recordings (306 channels, 1000 Hz) from nearly 700 subjects performing various tasks and in a resting state, collected using an Elekta-Neuromag system.

- **Go-Nogo**[16]: The Go-Nogo dataset includes EEG recordings (32 channels, 1000 Hz) from 14 subjects performing an animal categorization task and a recognition task, collected using a Neuroscan 5083 system.

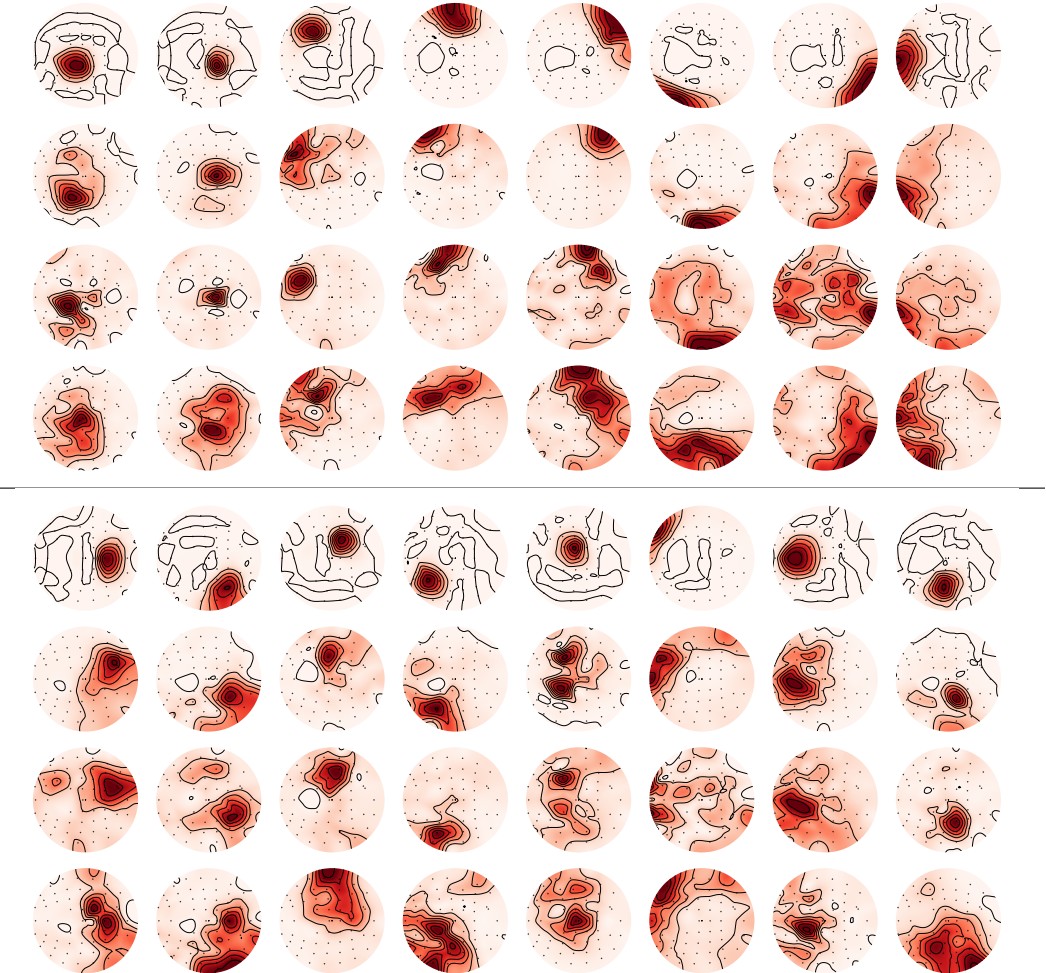

Figure 6: Visualisation of the topographic maps of cross-attention for each channel in BrainTokenizer. The model's 16 queries are displayed in 8 columns on the top and bottom, with the first eight and the last eight queries shown separately. The 4 rows represent the attention weights of each head in the multi-head cross-attention mechanism.

- **MusicEEG**[13]: The MusicEEG dataset includes EEG recordings (19 channels, 1000 Hz) from 31 subjects listening to 40 music clips of 12 s duration each, targeting a range of emotional states, collected using a BrainProducts BrainAmp system.

- **HFO**[11]: The HFO dataset includes sleep EEG recordings (23 channels, 1024 Hz) from 30 children and adolescents with focal or generalized epilepsy, collected using a Micromed EEG system.

- **AversiveMEG**[82]: The AversiveMEG dataset includes MEG recordings (275 channels, 1200 Hz) from 28 subjects completing an aversive learning task, collected using a CTF Omega system.

- **SRM**[27]: The SRM dataset includes EEG recordings (64 channels, 1024 Hz) from 111 subjects in resting state, collected using a BioSemi ActiveTwo system.

- **MIND**[81]: The MIND dataset includes MEG recordings (306 channels, 1250 Hz) from 8 subjects who received nerve electrical stimuli, collected using an Elekta-Neuromag system.

- **RestCog**[80]: The RestCog dataset includes EEG recordings (64 channels, 500 Hz) from 60 subjects during 3 experimental sessions together with sleep, emotion, mental health, and mind-wandering related measures, collected using a Brain Products GmbH system.

- **SMN4Lang**[79]: The SMN4Lang dataset includes MEG recordings (318 channels, 1000 Hz) from 12 subjects listening to 6 hours of naturalistic stories, collected using an Elekta-Neuromag system.

- **HBN EO/EC**[41]: The HBN EO/EC dataset includes EEG recordings (129 channels, 500 Hz) from 2952 children in resting state with eyes-open for 20 seconds and eyes-closed for 40 seconds, collected using an EGI system.

- **THINGS-MEG**[30]: The THINGS-MEG dataset includes MEG recordings (275 channels, 1200 Hz) of 4 subjects who were shown 22448 unique images of 1854 objects, collected using a CTF 275 MEG system.

- **ASWR-MEG**[21]: The ASWR-MEG dataset includes MEG recordings (160 channels, 1000 Hz) from 24 subjects listening to random word sequences and judging whether a probe word is related to the preceding word, collected using an Elekta-Neuromag system.

- **ImageLine**[68]: The ImageLine dataset includes MEG recordings (306 channels, 1000 Hz) from 30 subjects watching images of objects depicted as photographs, line drawings, or sketch-like drawings, collected using an Elekta-Neuromag system.

- **Features-EEG**[23]: The Features-EEG dataset includes EEG recordings (128 channels, 1000 Hz) from 16 subjects watching 256 oriented gratings that varied on four feature dimensions, collected using a BrainVision ActiChamp EEG system.

- **PEARL-Neuro**[51]: The PEARL-Neuro dataset includes EEG recordings (128 channels, 1000 Hz) from 79 subjects during resting state (eyes opened and closed) and cognitive tasks, including the multi-source interference task and Sternberg's memory task, collected using a Brain Products GmbH system.

- **NeuroMorph**[57]: The NeuroMorph dataset includes MEG recordings (208 channels, 1000 Hz) from 24 subjects doing a lexical decision task and a localizing task, collected using a KIT/Yokogawa MEG system.

- **Kymata-SOTO**[87]: The Kymata-SOTO dataset includes MEG recordings (306 channels, 1000 Hz) and EEG recordings (64 channels, 1000 Hz) from 20 subjects listening to English conversations and from 15 subjects listening to Russian conversations, collected using an Elekta-Neuromag system.

- **HBN-EEG**[60, 61, 62, 63, 64, 65, 66, 67]: The HBN-EEG dataset includes EEG recordings (128 channels, 500 Hz) from 1897 subjects who did 6 tasks including resting state, surround suppression, movie watching, contrast change detection, sequence learning, and symbol search, collected using a Magstim-EGI system.

- **Awakening**[6]: The Awakening dataset includes EEG recordings (65 channels, 5000 Hz) from 21 subjects during resting state and propofol sedation, aiming to investigate the effects of propofol on dreaming, collected using a Brain Products GmbH system.

# I  Downstream Finetuning Dataset Description

Below, we provide detailed descriptions of the downstream datasets used for finetuning BrainOmni.

- **Gloups-MEG**[76]: The Gloups-MEG dataset includes MEG recordings (248 channels, 2034.5 Hz) of 17 subjects completing a learning task and a resting-state condition, collected using an Elekta-Neuromag system.

- **PerceiveImagine**[6]: The PerceiveImagine dataset includes EEG recordings (64 channels, 1000 Hz) of 52 subjects watching an image for 6 seconds, and then imagining the image they see for 6 seconds, collected using a Neuroscan Synamps2 system.

- **TUAB**[47]: The TUAB dataset includes EEG recordings (21 channels, 256 Hz) from 2328 patients, annotated as normal or abnormal. A total of 408853 10-second samples were used for classification to predict these labels.

- **TUEV**[47]: The TUEV dataset includes EEG recordings (21 channels, 256 Hz) from 294 subjects, which are segmented into 112237 5-second samples across 6 classes: (1) spike and sharp wave (SPSW), (2) generalized periodic epileptiform discharges (GPED), (3) periodic lateralized epileptiform discharges (PLED), (4) eye movement (EYEM), (5) artifact (ARTF), and (6) background (BCKG). We perform a classification to predict these event labels.

- **MDD**[44]: The MDD dataset includes EEG recordings (20 channels, 256 Hz) from 35 patients with major depressive disorder and 30 normal controls across three sessions: eyes open, eyes closed, and task. A total of 7302 10-second samples were used for classification to predict the presence of major depressive disorder.

- **WBCIC_SHU**[83]: The WBCIC_SHU dataset includes EEG recordings (58 channels, 1000 Hz) from 51 subjects performing motor imagery tasks: left-hand grasping, right-hand grasping. A total of 30591 4-second samples were used for classification to predict the type of task.

- **PhysioNet-MI**[58]: The PhysioNet-MI dataset includes EEG recordings (64 channels, 160 Hz) from 109 subjects performing 4 motor imagery and movement tasks. A total of 9837 4-second samples were used for classification to predict the type of task.

- **FACED**[9]: The FACED dataset includes EEG recordings (30 channels, 1000 or 250 Hz) from 123 subjects watching 28 emotion-elicitation video clips covering 9 emotion categories. The coarse categories (negative, positive and neutral) are utilised. A total of 10332 10-second samples were used for classification to predict the emotion categories.

- **PD31**[56]: The PD31 dataset includes EEG recordings (32 channels, 512 Hz) from 16 healthy subjects and 15 subjects with Parkinson's disease during resting state. A total of 882 10-second samples were used for classification to predict the presence of Parkinson's disease.

- **ASD74**[19]: The ASD74 dataset includes MEG recordings (306 channels, 1000 Hz) from 35 children with autism spectrum disorders (ASD) and 39 typically developing children, who watched videos while receiving auditory stimuli. A total of 12320 10-second samples were used for classification to predict the presence of autism spectrum disorders.

- **SomatoMotor**[37]: The SomatoMotor dataset includes both EEG (74 channels, 1004 Hz) and MEG recordings (306 channels, 1004 Hz) from 5 subjects who received nerve electrical stimuli at the right wrist. The task was to lift the left index finger as quickly as possible after each right median nerve stimulus. A total of 1208 2-second samples were used for classification to predict whether their fingers were lifted because of nerve stimulus, or spontaneously.

- **AD65**[43]: The AD65 dataset includes EEG recordings (19 channels, 500 Hz) from 36 subjects with Alzheimer's disease and 29 healthy subjects during resting state. A total of 5349 10-second samples were used for classification to predict the presence of Alzheimer's disease.

- **MEG-MMI**[40]: The MEG-MMI dataset includes MEG recordings (269 magnetometer channels, 1200 Hz) from 29 adolescents with major depression and 22 healthy subjects during mood induction tasks. A total of 1770 30-second samples were used for classification to predict the presence of major depression.

## J Baseline Models Description

We compare BrainOmni with both pretrained and non-pretrained baseline models on various downstream tasks. The basic information of the baseline models is as follows:

- **CNN-Transformer**[52]: CNN-Transformer is a non-pretrained neural network that combines CNNs and transformer blocks to model long-range dependencies in CNN-derived features.

- **ContraWR**[85]: ContraWR is a non-pretrained neural network that consists of a cascaded pipeline beginning with a short-time Fourier transform (STFT), followed by a 2D CNN layer and three subsequent 2D convolutional blocks.

- **SPaRCNet**[35]: SPaRCNet is a non-pretrained 1D CNN based neural network with dense residual connections.

- **ST-Transformer**[70]: ST-Transformer is a non-pretrained transformer-based model that leverages the attention mechanism to better utilize both spatial and temporal features in EEG data.

- **FAMED**[31]: FAMED is a non-pretrained model for MEG-based epilepsy analysis, built on SCSE-ResNet with dilated convolutions.

- **LaBraM**[33]: LaBraM is a foundational EEG model pretrained on diverse datasets. It employs vector-quantized neural tokenization and masked channel prediction to effectively learn robust EEG features.

- **CBraMod**[78]: CBraMod is a foundational EEG model pretrained on diverse datasets. It employs a Criss-Cross Transformer architecture to separately model spatial and temporal dependencies, thereby effectively learning robust EEG features.

# K   Hyperparameters for model

This section presents the model parameters of BrainTokenizer and BrainOmni, as well as the training hyperparameters during pre-training and downstream fine-tuning.

Table 10: Hyperparameters for BrainTokenizer training

| | **Hyperparameters** | **Values** |
|---|---|---|
| | Window length | 512 |
| | N filters | 32 |
| | Ratios | [8, 4, 2] |
| | Kernel size | 5 |
| | Last Kernel size | 5 |
| | Hidden dim | 256 |
| Brain Tokeniser | Codebook dim | 256 |
| | Codebook size | 512 |
| | Num quantizers | 4 |
| | Rotation trick | True |
| | Latent source number | 16 |
| | Attention head number | 4 |
| | Dropout | 0.0 |
| Total batch per update | | 512 |
| Weight decay | | 1e-2 |
| Lr | | 2e-4 |
| Epochs | | 16 |
| | Type | AdamW |
| Optimizer | Betas | [0.5, 0.9] |
| | Eps | 1e-5 |
| | Type | WarmupCosineLR |
| Scheduler | Warmup ratio | 0.1 |
| | Cos min ratio | 0.05 |

Table 11: Hyperparameters for BrainOmni training

| Hyperparameters | | BrainOmni-tiny | BrainOmni-base |
|---|---|---|---|
| BrainOmni | Hidden dim | 256 | 512 |
| | Attention head number | 8 | 16 |
| | Attention depth | 12 | 12 |
| | Lr | 5e-4 | 4e-4 |
| | Overlap ratio | 0.25 | |
| | Dropout | 0.1 | |
| | Mask ratio | 0.5 | |
| Total batch per update | | 256 | |
| Epochs | | 32 | |
| Weight decay | | 5e-2 | |
| Optimizer | Type | AdamW | |
| | Betas | [0.9, 0.95] | |
| | Eps | 1e-6 | |
| Scheduler | Type | WarmupCosineLR | |
| | Warmup ratio | 0.1 | |
| | Cos min ratio | 0.1 | |

Table 12: Hyperparameters for downstream finetuning

| Hyperparameters | | Values |
|---|---|---|
| Total batch per update | | 128 |
| Epochs | | 30 |
| Weight decay | | 5e-2 |
| Label smoothing | | 0.1 |
| Optimizer | Type | AdamW |
| | Betas | [0.9, 0.99] |
| | Eps | 1e-6 |
| Scheduler | Type | WarmupCosineLR |
| | Warmup ratio | 0.1 |
| | Cos min ratio | 0.1 |

