# OpenReview forum: "BrainOmni: A Brain Foundation Model for Unified EEG and MEG Signals"
_NeurIPS.cc/2025/Conference — NeurIPS 2025 poster_

### Official Review · Reviewer_VHcX · 2025-07-01

**Clarity:** 2
**Significance:** 3
**Originality:** 2
**Rating:** 3
**Confidence:** 4

**Summary:**

This paper proposes BrainOmni, a foundational model for unified representation learning from EEG and MEG signals. It introduces BrainTokenizer, which transforms raw brain signals into discrete tokens by incorporating sensor metadata, enabling robust generalization across diverse devices. The model benefits from joint EEG-MEG pretraining and demonstrates strong performance on downstream tasks.

**Questions:**

1. Why was a 25% channel dropout rate selected for the BrainTokenizer? It would be helpful to know whether this value was determined through ablation studies or parameter tuning.

2. What is the individual contribution of each of the four loss functions used in training the BrainTokenizer (see Eq. 5)? A more detailed breakdown or ablation analysis of how each component of the loss function contributes to the overall training performance would strengthen the theoretical foundation of the model.

**Ethical Concerns:**

["NO or VERY MINOR ethics concerns only"]

**Final Justification:**

Most of my concerns have been addressed.

**Limitations:**

yes

**Quality:**

2

**Strengths And Weaknesses:**

**Strengths**

1. BrainOmni is the first electroencephalographic (EEG) foundation model that integrates both EEG and MEG modalities. The authors propose a novel Sensor Encoder that effectively fuses inputs from different modality devices. This contributes insight into the development of multimodal physiological signal foundation models.

2. The authors have curated a large-scale MEG dataset for pre-training.

3. The paper provides detailed descriptions of the pre-training, fine-tuning procedures, and model architecture,  significantly enhancing the reproducibility of the work.

**Weaknesses**

1. While the paper reports using 12 pre-training MEG datasets, only a single downstream MEG dataset is used for evaluation. The authors could strengthen the empirical validation by withholding 1–2 of the pre-training datasets for fine-tuning and evaluation, thereby providing a more realistic assessment of BrainOmni’s generalization capability on MEG-specific tasks.

2. The comparison methods selected for the MEG modality are primarily designed for EEG signals, which introduces a potential bias and leads to an unfair comparison (see Table 3). A more principled set of baselines specifically tailored for MEG would be necessary to draw solid conclusions.

3. The use of a cross-subject 3-fold validation setup raises concerns, especially since it deviates from the data splitting protocols used in prior foundation model studies. It would be more appropriate to adopt the same data partitioning scheme as existing FM models for fair comparison. Under the current setup, the performance of baseline foundation models appears to drop significantly—even when the implementation of these baselines is identical to that of competing methods like CbraMod—suggesting that the results may be sensitive to the chosen splits.

4. Fig. 3(c) shows the trend of latent source variables with respect to reconstruction loss, but the paper lacks analysis of their impact on the overall performance of BrainOmni. As the latent source variables influence the size of the tokens generated by the BrainTokenizer, they likely have a direct effect on the subsequent training of BrainOmni. The authors should provide a more in-depth discussion of this relationship and its implications.

---

> ### Author Rebuttal · Authors · 2025-07-31
>
> Thank you for your detailed feedback. We respond to your comments below:
>
> ### Regarding downstream MEG datasets
>
> We further introduce an additional publicly available MEG dataset, MEG-MMI \[1\]. The results (in balanced accuracy) are listed below:
> |  | CNNTransformer | ContraWR      | SPaRCNet      | STTransformer | BrainOmni     |
> |--------|----------------|---------------|---------------|---------------|---------------|
> | BACC   | 0.513 ± 0.019  | 0.496 ± 0.005 | 0.616 ± 0.080 | 0.610 ± 0.020 | 0.663 ± 0.052 |
>
> It can be seen that BrainOmni still outperforms the baselines on the additional MEG dataset.
>
> \[1\] Liuzzi, Lucrezia, et al. "Magnetoencephalographic correlates of mood and reward dynamics in human adolescents." _Cerebral cortex_ 32.15 (2022): 3318-3330.
>
> ### Regarding MEG baselines
>
> We would like to first clarify that all of our chosen baselines are not specifically designed for EEG but rather general-purpose architectures for multi-channel time-series data -- spanning CNN+Transformer (CNNTransformer), pure CNN (ContraWR and SPaRCNet), and separately spatial and temporal Transformer (STTransformer). Training them directly on MEG, therefore, provides a fair indication of how those paradigms perform in the MEG domain. We further include an additional model, FAMED \[2\], primarily designed for MEG signals. The results (in balanced accuracy) are shown in the following table:
>
> | Dataset  | CNNTransformer  | ContraWR       | SPaRCNet       | STTransformer  | FAMED          | BrainOmni_tiny |
> |----------|-----------------|----------------|----------------|----------------|----------------|----------------|
> | asd74    | 0.508 ± 0.011   | 0.533 ± 0.047  | 0.565 ± 0.039  | 0.532 ± 0.024  | 0.566 ± 0.038  | 0.606 ± 0.019   |
> | MEG‑MMI  | 0.513 ± 0.019   | 0.496 ± 0.005  | 0.616 ± 0.080  | 0.610 ± 0.020  | 0.505 ± 0.010  | 0.663 ± 0.052   |
>
> Results show the superior performance of BrainOmni.
>
> \[2\] Hirano, Ryoji, et al. "Deep learning based automatic detection and dipole estimation of epileptic discharges in MEG: a multi-center study." _Scientific Reports_ 14.1 (2024): 24574.
>
> ### Regarding cross-subject 3-fold validation setup
>
> In the downstream part, we evaluated a variety models on a variety of datasets. Since most datasets lack standard train/test splits (only TUAB and TUEV have) and some compared models report results using different or undisclosed splits (_e.g._, AD65 dataset used in \[3\]), we built a unified evaluation pipeline across all datasets and models to guarantee fully comparable results. The cross-subject 3-fold validation setup, where every individual appears once in test, can capture inter-subject variability better than a single fixed split, achieving a more comprehensive and robust assessment.
>
> To verify that our performance is not an artifact of this protocol, we followed CbraMod’s original PhysioNet-MI dataset filtering (only use motor imagery sessions) and splitting (subject (1-70, 71-89, 90-109) for (train, valid, test)) procedure, repeated experiments with 5 random seeds, and report the averaged metrics of BrainOmni. Results are shown in the following table:
>
> | Model | Setting |Balance Accuracy|Cohen's Kappa    |Weighted F1      |
> |-------|----------|----------------|-----------------|-----------------|
> |CBraMod | result in \[4\] | 0.642 ± 0.009   |0.522 ± 0.017    |0.643 ± 0.010    |
> |BrainOmni | same split as \[4\] | 0.650 ± 0.006   | 0.533 ± 0.007    | 0.651 ± 0.005    |
> |CBraMod | reshuffled split | 0.572 ± 0.008   | 0.430 ± 0.011    | 0.574 ± 0.008    |
>
> Comparing row 1 and 2, BrainOmni still outperform competing methods under the same splitting, demonstrating that the gains are not due to differences in data assignment.
>
> Furthermore, we generated an alternative subject split, by reshuffling the subject list (with random seed 0) while maintaining the train/val/test ratio as \[4\]. Then we reproduced CBraMod's downstream experiments strictly using their codebase and following their hyperparameters listed in \[4\]. The results are listed in row 3 of the above table.
>
> It can be observed that the baseline performance dropped significantly, demonstrating the sensitivity caused by inter-subject variability, and highlighting the necessity and robustness of our cross-subject 3-fold setup.
>
> \[3\] Yuan, Zhizhang, et al. "Brainwave: A brain signal foundation model for clinical applications." _arXiv preprint arXiv:2402.10251_ (2024).
>
> \[4\] Wang, Jiquan, et al. "CBraMod: A Criss-Cross Brain Foundation Model for EEG Decoding." _The Thirteenth International Conference on Learning Representations_ (2025).
>
>
> ### Regarding the number of latent source variables
>
> We added an ablation study on the number of latent source variables $ Q \in \{8, 16, 32\} $, with results listed below:
>
> | Num of latent source variables | TUEV (EEG)    | asd74 (MEG)   | SomatoMotor (EMEG) |
> | ------------------------------------ | ------------- | ------------- | ------------------ |
> | 8 | 0.590 ± 0.017 | 0.558±0.019 | 0.773±0.140 |
> | 16 | 0.597 ± 0.019 | 0.606 ± 0.019 | 0.787 ± 0.112      |
> | 32 | 0.605 ± 0.022 | 0.613±0.036 | 0.776±0.116 |
>
> Overall, Q=8 performs worse than Q=16 and Q=32, while Q=16 and Q=32 show similar performance across three datasets. Combined with the trend in Figure 3\(c\), this shows that too few source variables lead to poor reconstruction and limited downstream performance, whereas increasing Q beyond 16 brings diminishing improvements.
>
> ### Regarding the channel dropout rate
>
> The channel dropout serves as a structural regulariser, enabling BrainTokenizer to adapt to varying channel layouts and quantities, as well as equipping it with the capability to predict target position electrodes based on existing ones, which improves robustness to channel corruption or occlusion. The 25% value was chosen based on experience. We did not perform a full hyperparameter search on this value due to computational constraints, but we expect that tuning this parameter could further improve results.
>
> ### Regarding the loss functions in BrainTokenizer
>
> The BrainTokenizer training loss contains four components: time-domain loss, frequency-domain loss, Pearson correlation loss, and RVQ commitment loss. Given that the time-domain and commitment losses are essential for VQ-AE training, we add an ablation study for how frequency loss and PCC loss influence the reconstruction performance, with results listed in the following table.
>
> | time & commitment | freq loss | PCC loss       |MAE $\downarrow$ |AMP $\downarrow$|PHASE $\downarrow$|PCC value $\uparrow$ |commitment $\downarrow$|total loss $\downarrow$|
> |-|-|---------|---|---|-----|---------|----------|----------|
> | &#10004; | |  |0.799|0.256|2.154|0.002|5.01e-7|4.207|
> | &#10004; | &#10004; |  |0.471|0.160|1.583|0.716|1.64e-4|2.517|
> | &#10004; | | &#10004; |0.462|0.167|1.863|0.751|1.83e-4|2.703|
> | &#10004; | &#10004; | &#10004; |0.480|0.165|1.495|0.723|1.48e-4|2.626|
>
> With only time-domain and commitment loss, the decoder outputs collapsed into near‑constant signals. Introducing the frequency‑domain loss greatly improved spectral detail (lower MAE/AMP/PHASE), but led to spurious high‑frequency impulses. The PCC term mitigates these impulses by aligning overall waveform trends, yet on its own provides weaker spectral reconstruction. Finally, we chose to employ both loss functions simultaneously, suppressing the impulse while preserving as much frequency-domain detail as possible.
>
> We sincerely appreciate your feedback and hope that our responses adequately address your concerns.

---

> > ### Comment · Reviewer_VHcX · 2025-08-03
> >
> > I appreciate the authors detailed response. Most of my concerns have been adequately addressed. I will adjust my score accordingly.

---

> > > ### Author Response · Authors · 2025-08-04
> > >
> > > We are glad to have addressed your concern and sincerely appreciate your adjustment of the score. Thank you again for your insightful suggestions, which will undoubtedly help us further improve our paper.

---

### Official Review · Reviewer_ThgG · 2025-07-01

**Clarity:** 3
**Significance:** 3
**Originality:** 3
**Rating:** 5
**Confidence:** 4

**Summary:**

This paper proposes BrainOmni, a foundation model designed to learn unified representations from both EEG and MEG brain signals. The authors introduce a BrainTokenizer module that converts raw neural recordings—along with sensor-specific metadata—into a shared discrete token representation, enabling compatibility across different devices and modalities. A Criss-Cross Transformer is then employed to capture spatiotemporal dependencies among the tokens and generate contextualized embeddings. The model is rigorously evaluated across a wide range of downstream tasks, where the learned BrainOmni embeddings consistently outperform those produced by existing brain foundation models.

**Questions:**

Is the reconstruction visualization (in main and in supplementary) before or after finetuning? Have you explored the effect of finetuning on Tokenizer and the transformers separately?

**Ethical Concerns:**

["NO or VERY MINOR ethics concerns only"]

**Final Justification:**

This is a solid paper.

**Limitations:**

yes

**Quality:**

3

**Strengths And Weaknesses:**

One key strength of this paper lies in itsimpactful research direction. Developing a foundation model capable of integrating heterogeneous neural data across EEG and MEG modalities addresses a long-standing bottleneck in neuroscience: the fragmentation of datasets and the scarcity of large-scale annotated data for individual tasks. By introducing a unified representation framework, BrainOmni potentially offers a scalable solution to harmonize recordings from diverse sources, devices, and paradigms.

Another notable strength is the rigor and comprehensiveness of the experimental validation. The model is evaluated across 10 diverse downstream tasks, covering a wide range of cognitive, clinical, and motor domains. The authors conduct extensive ablation studies to isolate the contributions of each architectural component. Further, the paper includes additional analyses to probe the learned network, which adds certain level of transparency to the model’s behavior. The authors also commit to open-sourcing the code and pretrained models, which promotes reproducibility.

One notable weakness of this paper is its inferior performance on several benchmark datasets compared to task-specific, fine-tuned models. For example, on the PD31 dataset, classical methods using carefully engineered features achieve over 95% accuracy [1], whereas BrainOmni reports only 73%. Similar performance gaps can be found across other tasks in the literature. While BrainOmni demonstrates competitive results relative to other foundation models, it fails to match the state-of-the-art performance achieved by specialized models tailored to specific domains. Admittedly, BrainOmni offers superior generalizability across modalities and datasets, which is a key advantage of foundation models. However, the authors should have cited and compared against these specialized baselines, and provided a balanced discussion of BrainOmni’s strengths and limitations in relation to them. A forward-looking perspective on how brain foundation models can offer practical value, even when downstream accuracy is lower, would strengthen the impact and positioning of this work.

[1] Aljalal, Majid, et al. "Detection of Parkinson’s disease from EEG signals using discrete wavelet transform, different entropy measures, and machine learning techniques." Scientific Reports 12.1 (2022): 22547.

---

> ### Author Rebuttal · Authors · 2025-07-31
>
> Thank you for your insightful feedback and for acknowledging our research direction and experimental validation. We response to your comments below:
>
> ### Regarding the performance compared to specialised models
>
> We appreciate you for raising this important point, and agree on the importance of explicitly addressing these points. Paper \[1\] designs channel combinations and feature extraction methods based on the distribution of features across frequency bands and brain regions, demonstrating the outstanding performance of machine learning approaches in this task. That said, we would like to highlight a key difference in experimental setup and data splitting strategy, which makes direct comparison of results between BrainOmni and the cited classical methods inappropriate for our context.
>
> The referenced baseline (\[1\]) employs a **random split** approach on the PD31 dataset, which may result in the same subjects appearing in both training and testing sets, thereby compromising subject independence. Instead, our experiments utilise a **cross-subject split**, ensuring no subject-level information leakage, which is a more challenging and practically relevant evaluation setting. Indeed, paper \[1\] explicitly acknowledges concerns regarding data leakage associated with random splits in the "UNM‑based results" section.
>
> Since the implementation of [1] is not open-sourced and the specific data split used is not disclosed, a direct comparison under a unified setup is not feasible. To estimate the impact of the split strategy, we conducted an additional experiment using a random split on PD31, following a similar protocol. This resulted in a substantially higher accuracy of 92% for BrainOmni, highlighting that the choice of data split can significantly affect performance.
>
> We fully agree with your suggestion that it is important to provide a balanced discussion of BrainOmni’s strengths and limitations in relation to specialised baselines. This also reflects a broader challenge in current AI research: general-purpose models often trade off some task-specific performance in exchange for scalability, flexibility, and cross-domain transferability.
>
> BrainOmni is designed to serve as a unified foundation model across heterogeneous non-invasive neurophysiological electrophysiological modalities (specifically EEG and MEG). It supports cross‑device and cross‑subject generalisation, making it suitable for practical applications where domain shifts are inevitable. While it may not always outperform domain-specific baselines tailored to one dataset or one task, it offers broader coverage and generalisability across datasets, tasks, and modalities — a key advantage for scalable and reusable neuro-AI systems. We will add this discussion to our manuscript.
>
> \[1\] Aljalal, Majid, et al. "Detection of Parkinson’s disease from EEG signals using discrete wavelet transform, different entropy measures, and machine learning techniques." _Scientific Reports_ 12.1 (2022): 22547.
>
> ### Regarding the questions
>
> Answer to Q1: We visualise the reconstruction results of unseen devices without finetuning.
>
> Answer to Q2: We didn't separately finetune the BrainTokenizer because it relies on a learned VQ codebook whose discrete embeddings encode meaningful spatiotemporal tokens, and updating it on limited downstream data risks corrupting those codebook semantics. Instead, following standard practice, we freeze the entire Tokeniser (including the VQ quantiser) during downstream adaptation and finetune only the Transformer layers.
>
> Thank you again for your positive comments and constructive suggestions. We hope our response resolves your concerns.

---

### Official Review · Reviewer_5FEv · 2025-07-02

**Clarity:** 2
**Significance:** 4
**Originality:** 3
**Rating:** 5
**Confidence:** 4

**Summary:**

The paper describes a foundation model that is trained to ingest either EEG or MEG brain data. The model consists of a raw data tokenizer (BrainTokenizer) and a Transformer encoder (BrainOmni). The tokenizer is trained as a masked autoencoder, and comprises a sensor encoding scheme to model sensor characteristics such as position, orientation and type. The quantized tokens are used as input to a Transformer encoder which is trained with a masked token prediction task. Both parts are pretrained on a corpus of 1997 hours of EEG and 656 hours of MEG. The model is then reused on 8 EEG downstream tasks and 2 MEG downstream tasks, and its performance is compared to fully supervised baselines and two existing foundation models. Results suggest that the proposed model outperforms baselines on most tasks, and ablation studies suggest that the joint training on both EEG and MEG yields improved performance.

**Questions:**

1. Can you describe the cross-validation/testing strategy in more detail, e.g. with pseudo-code? It is not clear to me whether a first train/test split is done, followed by a 3-fold cross-validation on the training set, or if the 3-fold CV is done over the entire dataset.
2. How can we be sure that the effect of joint pretraining (results of Table 5) is not mainly a matter of pretraining on a lot more data? It looks like the improvement over the MEG-only model is larger than for the EEG-only model, which could support this interpretation as there is about 3x more EEG data in the pretraining set.
3. Can you provide more information on the “power-spectral-density-based detection algorithm”? Similarly, can you give more details into what are “reference inconsistencies” (line 184) and how reference signals are used in cases where signals have already been referenced and the reference channel is not available? Overall, I wonder whether the BrainTokenizer might be able to handle these steps directly. Finally, can you confirm whether the bad channel detection is done at the recording- or segment-level?

**Ethical Concerns:**

["NO or VERY MINOR ethics concerns only"]

**Final Justification:**

The submission presents convincing evidence that the proposed brain foundation model competes with existing models and that joint training with EEG and MEG yields improved performance. My comments have been appropriately addressed during the rebuttal discussion.

**Limitations:**

Yes.

**Quality:**

3

**Strengths And Weaknesses:**

Strengths

* Quality: The submission is technically sound, with claims that are overall well supported by experimental results (evaluation on 9 downstream tasks, testing cross-device generalization on unseen devices, and relevant ablation studies).
* Clarity: The paper is overall clearly written and easy to follow, though some important information appears to be missing (see Weaknesses).
* Significance: The results are likely to positively impact the brain foundation model community, as it combines two common modalities (EEG and MEG) that are typically studied independently. It also provides a new architecture/pretraining recipe with an interesting connection to the backward/forward models used in traditional biophysical modelling of M/EEG. Finally, the comparison to existing baselines suggest the proposed model outperforms the state of the art on most downstream tasks.
* Originality: The paper presents the first model to meaningfully combine MEG and EEG to build foundation models, along with interesting innovations for the handling of varying channel layouts.

Weaknesses

* Clarity: Some relevant information appears to be missing from the manuscript:
    * In the channel compression step of Section 2.2, it is unclear what the sequence length is for the input of the cross-attention layer (C x W, or only C), and therefore whether the module attends to all timesteps or if each timestep is treated independently.
    * In the description of the downstream evaluation (Section 3.2): it is not clear how the train/test sets are built, i.e. whether this is done as part of the 3-fold CV split, or if all datasets already provide a fixed test set. I assume that the 3-fold CV is carried out on a training set, but it is not obvious from the text.
    * Finally, it is unclear how the pretrained BrainOmni is finetuned, i.e. whether a linear projector or MLP is used on top to match the output dimensionality.
    *I didn’t find the parameter count for the BrainTokenizer in the manuscript.
* Quality: Depending on how the downstream evaluation was actually carried out, a common optimization configuration (i.e. training for 30 epochs) for all downstream tasks and models may be yielding more severe overfitting in smaller models.

---

> ### Author Rebuttal · Authors · 2025-07-31
>
> Thank you for your positive comments and thoughtful feedback. We response to your comments below:
>
> ### Regarding the sequence length in channel compression step
>
> The sequence length fed into the cross-attention layer is **only C** (the number of channels). The channel compression module does a cross-attention operation to project heterogeneous inputs into a unified latent source variable. The learnable query is shared across all windows, ensuring that channel compression remains consistent between different temporal windows. Local temporal feature is captured by the SEANet Encoder, and long-range spatiotemporal dependencies are subsequently modelled in Stage 2 by the Criss‑Cross Transformer.
>
> ### Regarding the 3-fold CV strategy
>
> Since most datasets lack standard train/test splits and some compared models report results using different or undisclosed splits, we chose to use the 3-fold cross-validation to ensure a fair comparison while also taking computational cost into account. Specifically, for datasets without an official split (_i.e.,_ datasets except TUAB and TUEV), we first split all subjects into three equally sized subsets A, B and C. Then three experiments are performed in which (train, validation, test) are assigned in turn to (A, B, C), (B, C, A) and (C, A, B). For TUAB and TUEV, which each provide an official train/eval split, we keep the eval set for all testing, split the official training subjects equally into three folds, and in each of the three runs use two folds for training and the remaining fold for validation.
>
> ### Regarding the finetuning strategy
>
> In downstream fine-tuning, the pretrained BrainOmni produce embedding in shape `[B, Q, W, D]` (B - batch size, Q - number of compressed channels, W - temporal window length, and D - embedding dimension). The embedding is firstly average-pooled over W, and then flattened to `[B, Q*D]`. The classification head is a two-layer MLP. The BrainTokenizer has 5M parameters.
>
> As for the number of training epochs, we would like to clarify that rather than using the 30‑epoch checkpoint for testing, we evaluated model performance on the held‑out validation set at the end of each epoch and used the checkpoint that achieved the highest balanced accuracy on the validation set for final test evaluation. This strategy prevents smaller models from overfitting due to excessive training epochs.
>
>
> ### Regarding the effect of joint training
>
> Firstly, we would like to highlight that a key goal of BrainOmni is to scale across EEG and MEG modalities, especially given MEG's complexity and limited data availability. Although joint training on both EEG and MEG data naturally increases the total amount of pretraining data, it is not guaranteed to help. In fact, unifying these two modalities in a single framework is non-trivial. Due to their distinct signal characteristics, spatial-temporal resolutions, and data formats, naïvely combining EEG and MEG may even hurt performance, as the model could divert capacity to modality-specific patterns, compromising optimal learning for either modality. Despite these challenges, BrainOmni successfully supports joint training across EEG and MEG: results in Table 5 show that EMEG pretraining improves downstream performance on both modalities.
>
> For further verification, we ran an ablation experiment in which we subsampled the combined EMEG pretraining set to match the size of the EEG‑only and MEG‑only corpora. The results are listed below:
>
> | Configuration                        | TUEV (EEG)    | asd74 (MEG)   | SomatoMotor (EMEG) |
> | ------------------------------------ | ------------- | ------------- | ------------------ |
> | MEG-only (650 hours)                 | –             | 0.530 ± 0.032 | 0.683 ± 0.168      |
> | EEG-only (2000 hours)                | 0.555 ± 0.003 | -             | 0.737 ± 0.042      |
> | Joint ratio 0.25 (close to MEG only) | 0.571 ± 0.009 | 0.569±0.024 | 0.780±0.172  |
> | Joint ratio 0.75 (close to EEG only) | 0.592 ± 0.003 | 0.614±0.011 | 0.775±0.115 |
> | Joint train                          | 0.597 ± 0.019 | 0.606 ± 0.019 | 0.787 ± 0.112      |
>
> Results above show that (i) given roughly the same amount of pretraining data, joint training using EEG and MEG helps improve the performance (comparing rows 1&2 to rows 3-5); (ii) an increase in the total amount of pretraining data does not necessarily yield improved performance (rows 3-5).
>
>
> ### Regarding the preprocessing
>
> The power-spectral-density-based bad-channel detection is implemented as follows:
> ```csharp=
> Input: raw_data, threshold=10
> 1. Compute per‑channel PSD over the full recording:
>      PSD ← compute_psd(raw_data).data
> 2. Stabilise and log‑transform:
>      L ← log(PSD + 1e‑16)
> 3. Compute pairwise distances between channel spectra:
>      for i in 1…C, j in 1…C:
>          dist[i,j] ← L[i] – L[j]
> 4. Mean distance per channel:
>      m[i] ← mean(j, dist[i,j])
> 5. Identify outliers via IQR:
>      Q1 ← 25th_percentile(m), Q3 ← 75th_percentile(m)
>      IQR ← Q3 – Q1
>      upper ← Q3 + threshold·IQR
>      lower ← Q1 – threshold·IQR
>      bad_channels ← { ch_i | m[i] > upper or m[i] < lower }
> ```
>
> The reference inconsistency is caused by the fact that different data recording setups may use different physical reference electrodes. To address this, we first compute a global average reference by taking the mean across all channels, and then subtract this channel-average signal from each channel waveform, effectively aligning all channels to the same virtual reference. This protocol is applied regardless of whether a reference channel is unavailable or the signals have already been referenced, and it is performed both at the recording level and separately for each signal type.
>
> Thank you again for your positive comments and constructive suggestions. We will further clarify the above points in our manuscript. We hope our response resolves your concerns.

---

> > ### Comment · Reviewer_5FEv · 2025-08-04
> >
> > Thank you to the authors for addressing my comments and providing additional information.
> >
> > Effect of joint training: can you confirm what the pretraining set size (in hours) for EEG and MEG for rows 3-5 is? Also, can you confirm the last column is the joint EMEG performance, and not the modality-specific performance (as in Table 5)?

---

> > > ### Author Response · Authors · 2025-08-05
> > > **Further clarification**
> > >
> > > Thank you for pointing this out. First, we sincerely apologise for a formatting error that occurred during the conversion of the original results to the markdown editor on the SomatoMotor dataset. This error inadvertently displayed the MEG-only results for the joint ratio models, rather than the intended joint EMEG results, in the last column. Below in the table, we show the correct results of EEG-only, MEG-only, and models trained with different joint ratios. We have carefully checked the results for the other two datasets (TUEV and asd74), and confirmed they were correctly reported.
> > >
> > >
> > > To further clarify the pretraining set size:
> > > * row 1: Only using MEG data (~650 hours)
> > > * row 2: Only using EEG data (~2000 hours)
> > > * row 3: We segmented the data into 30-second clips and uniformly sampled 25% of the total (~660 hours), which roughly matches the total amount of MEG data.
> > > * row 4: We segmented the data into 30-second clips and uniformly sampled 75% of the total (~1990 hours), which roughly matches the total amount of EEG data.
> > > * row 5: Using all EEG and MEG data (~2650 hours)
> > >
> > >
> > > |  | SomatoMotor (EEG test) | SomatoMotor (MEG test) | SomatoMotor (EMEG test) |
> > > | --- | --- | --- | --- |
> > > | MEG-only | - | 0.683±0.168 | - |
> > > | EEG-only | 0.737±0.042 | - | - |
> > > | Joint train (25\% set, close to the amount of MEG-only) | 0.718±0.118 | 0.780±0.172 | 0.773±0.041 |
> > > | Joint train (75\% set,close to the amount of EEG-only) | 0.749±0.066 | 0.775±0.115 | 0.778±0.107 |
> > > | Joint train (full set) | 0.747±0.088 | 0.786±0.121 | 0.787±0.112 |
> > >
> > > Despite the updated results, the main conclusions remain unchanged: (i) when using approximately the same amount of pretraining data, joint training with both EEG and MEG improves performance (as seen when comparing rows 1 and 2 to rows 3–5); (ii) increasing the total amount of pretraining data does not necessarily lead to better performance (rows 3–5).

---

> > > > ### Comment · Reviewer_5FEv · 2025-08-07
> > > >
> > > > Thank you for the updated results which indeed confirm the original interpretation. My concerns have been addressed.

---

### Official Review · Reviewer_73i4 · 2025-07-03

**Clarity:** 3
**Significance:** 3
**Originality:** 2
**Rating:** 4
**Confidence:** 3

**Summary:**

This paper presents BrainOmni, a brain foundation model to generalize across EEG and MEG signals. It uses 1997 hours of EEG and 656 hours of MEG data from publicly available sources for pretraining. Their approach consists of BrainTokenizer to infer spatio temporal patterns of brain activities and SensorEncoder to use physical sensor properties. The evaluation results show the approach is promising.

**Questions:**

- What are the computational costs for training and inference? Is BrainOmni feasible for clinical use or only research labs?
- How does the model generalize across subjects with different brain anatomy or signal quality?
- More in-depth ablation study would be helpful to understand the benefits of each component.

**Ethical Concerns:**

["NO or VERY MINOR ethics concerns only"]

**Final Justification:**

It's nice to see a model that explicitly supports multiple modalities of brain signals -- EEG and MEG. It puts together and utilizes 1997 hours EEG and 656 hours MEG from public data sources is commendable. The authors are strongly encouraged to release the final dataset and pre-training model along with the weights.

**Quality:**

3

**Strengths And Weaknesses:**

Strengths
- BrainOmni is a significant step toward a general-purpose cognitive decoding model by leveraging both EEG and MEG data.
- The model is used using 1997 hours EEG and 656 hours MEG data from public sources, the largest I've seen so far.

Weaknesses
- No clear discussion of training efficiency, hardware requirements, or inference speed, which is crucial for scaling.
- More ablation study on where are the benefits coming from (e.g., increased dataset, using EEG and MEG together, specific design of neural networks). These insights are valuable to the research community.

---

> ### Author Rebuttal · Authors · 2025-07-31
>
> Thank you for the comments and for acknowledging that BrainOmni is a step towards a general-purpose model. We would like to first clarify several important misunderstandings presented in the summary:
> - BrainOmni is designed for **EEG and MEG** signals, not "functional MRI", as stated in the title.
> - All datasets used in our study are listed in Appendices D and E, without including "NSD, HCP, or Movie10K fMRI datasets".
> - As shown in Table 1, our downstream tasks include clinical (e.g., Alzheimer’s, Parkinson’s), abnormal (e.g., TUAB), event (e.g., TUEV), emotion (FACED), and motor imagery tasks. None of them fall under the category of "cognitive decoding", and no "visual-language fMRI tasks" were involved.
>
> We then respond to each comments below:
>
> > **Weakness1:** No clear discussion of training efficiency, hardware requirements, or inference speed, which is crucial for scaling.
>
> Thank you for pointing this out. Our pre-training was performed on 2\*8 NVIDIA A100 GPUs. Stage 1 (BrainTokenizer) took approximately 11 hours. Stage 2 required about 14 hours for the tiny model and 18 hours for the base model. Time cost for downstream fine‑tuning depends on dataset size. On a single A100, it took roughly 60 samples/sec for training and 90 samples/sec for inference. We will add this information to our manuscript.
>
> >  **Weakness2:** Attention map visualisations are provided, but no in-depth explanation of specific learned cognitive representations or their neuroscientific implications.
>
> We discussed the neuroscience background and how BrainTokenizer parallels the classical source estimation process in Appendix B. Specifically, the cross‑attention between raw EMEG signals and the sensor layout implements the backwards (sensor -> source) and forward (source -> sensor) mappings, so that each latent "source" functionally mirrors a cortical current source. We presented the topographic attention map in Appendix J. It shows that each source variable in our model has automatically learned a hierarchical structure, indicating that each source variable focuses on extracting features from specific scalp regions. This supports our hypothesis that BrainOmni learns meaningful, brain‑relevant representations rather than generic signal features.
>
> > **Weakness3:** While diverse in modality, most benchmarks are still based on visual-language fMRI tasks.
>
> As stated in the clarification above, this study focuses on EEG and MEG signals and does not involve visual-language fMRI tasks.
>
> > **Weakness4:** It’s unclear how well BrainOmni generalizes across individuals, or whether subject-specific adaptation is needed.
>
> We would like to highlight that all downstream evaluations are conducted in a **strict cross‑subject** setup, with no overlap of individuals between train, validation, and test splits. BrainOmni shows superior performance under this setting (Table 2), demonstrating its robust generalisation to unseen subjects.
>
> > **Question1:** What are the computational costs for training and inference? Is BrainOmni feasible for clinical use or only research labs?
>
> Please refer to the point 1 for the computational device and time costs. As for applicability, it's worth noting that BrainOmni-tiny has only ~10 million parameters, which is lightweight and suitable for clinical settings as well.
>
> > **Question2:** How does the model generalize across subjects with different brain anatomy or signal quality?
>
> Firstly, as stated in Section 2.2, we designed a Sensor Encoder which encodes the physical coordinates of each channel, enabling BrainOmni to cope with different device layouts across datasets and subjects. Secondly, we pretrained BrainOmni on over 2600 hours of EMEG data from diverse sources, improving generalisation across differing anatomies and noise levels. Preprocessing steps (bad channel detection, channel re‑referencing, normalisation, _etc._) also help to improve the robustness regarding signal quality.
>
> As stated in response to weakness 4, BrainOmni shows superior cross‑subject generalisation performance to unseen subjects. As discussed in Section 4.2, BrainOmni also shows robust cross-device generalisation to unseen devices.
>
> > **Question3:** Can you further interpret the learned representations or link them to known cognitive networks?
>
> Please refer to the response to Weakness 2 for a detailed explanation for the neuroscientific implications of learned representations.
>
> > **Question4:** Can BrainOmni be extended to handle EEG, MEG, or structural MRI for truly multi-modal brain modelling?
>
> As stated in the clarification, BrainOmni is designed as a foundation model for unified EEG and MEG signals. Table 5 demonstrates the effectiveness of this unified pretraining: joint E/MEG pretraining consistently outperforms single-modality pretraining across all datasets. This serves as a crucial first step toward broader multi-modal modelling,  which is inherently challenging due to the structural and functional differences across neural signal types. We look forward to extending BrainOmni to include other neural modalities, as discussed in Appendix A.
>
> We thank you again for the comments and we hope our rebuttal fully resolves your concerns.

---

> ### Author Response · Authors · 2025-08-08
>
> We thank Reviewer 73i4 for the comments. We believe we have addressed your concerns in detail in the rebuttal and hope our responses fully resolve the issues raised.

---

### Decision · Program_Chairs · 2025-09-17

**Decision:**

Accept (poster)

**Comment:**

This paper presents BrainOmni, a foundation model that unifies EEG and MEG signals through a BrainTokenizer module and SensorEncoder, enabling robust cross-modality and cross-device generalization. The model is pretrained on an unprecedentedly large corpus of 1997 hours of EEG and 656 hours of MEG data, and evaluated on 10 diverse downstream tasks. Results show that BrainOmni consistently outperforms existing foundation models and supervised baselines, with joint EEG–MEG pretraining yielding further gains.

The paper’s key strengths lie in its scale, novelty, and potential impact. Reviewers highlighted the significance of being the first foundation model to integrate EEG and MEG (73i4, 5FEv, VHcX), the thorough experimental validation across clinical, motor, and cognitive domains (ThgG), and the innovative design of BrainTokenizer, which draws inspiration from classical source estimation methods. The commitment to open-sourcing data and pretrained models further strengthens its value for the community.

The main weaknesses are the lack of detailed discussion on computational efficiency, the limited MEG downstream evaluation, and the absence of comparisons to some domain-specific baselines. Reviewers also requested more ablation studies, which the rebuttal addressed with extensive additional analyses. Clarifications on training/inference costs, ablations on loss components, and new MEG datasets helped resolve most concerns.

Overall, the contributions are timely and technically solid, offering a promising step toward general-purpose brain foundation models. I recommend acceptance.